# Drainage reorganization induces deviations in the scaling between valley width and drainage area

Elhanan Harel[1], Liran Goren[1], Onn Crouvi[2], Hanan Ginat[3], and Eitan Shelef[4].

[1]Earth and Environmental Sciences, Ben-Gurion University of the Negev, Beer Sheva, 84105, Israel.
[2]Geological Survey of Israel, Yesha'yahu Leibowitz 32, Jerusalem 9692100, Israel.
[3]The Dead-Sea and Arava Science Center, Tamar regional council Dead-Sea mobile post 86910, Tamar regional council, Israel.
[4]Geology and Environmental Science, University of Pittsburgh, 4107 O'Hara Street, Pittsburgh, Pennsylvania 15260- 3332, United States.

*Correspondence to*: Harel E. (elhananh@post.bgu.ac.il)

**Abstract**. The width of valleys and channels affects the hydrology, ecology, and geomorphic functionality of drainage networks. In many studies, the width of valleys and/or channels ($W$) is estimated as a power law function of the drainage area ($A$), $W = k_c A^d$. However, in fluvial systems that experience drainage reorganization, abrupt changes in drainage area distribution can result in valley or channel widths that are disproportional to their drainage areas. Such disproportionality may be more distinguished in valleys than in channels due to a longer adjustment timescale for valleys. Therefore, the valley width-area scaling in reorganized drainages is expected to deviate from that of drainages that did not experience reorganization.

To explore the effect of reorganization on valley width - drainage area scaling, we studied 12 valley sections in the Negev desert, Israel, categorized into undisturbed, beheaded, and reversed valleys. We found that the values of the drainage area exponents, *d,* are lower in the beheaded valleys relative to undisturbed valleys, but remain positive. Reversed valleys, in contrast, are characterized by negative *d* exponents, indicating valley narrowing with increasing drainage area. In the reversed category, we also explored the independent effect of channel slope ($S$) through the equation $W = k_b A^b S^c$ , which yielded negative and overall similar values for *b* and *c*.

A detailed study in one reversed valley section shows that the valley narrows downstream whereas the channel widens, suggesting that, as hypothesized, the channel width adjusts faster to post-reorganization drainage area distribution. The adjusted narrow channel dictates the width of formative flows in the reversed valley, which contrasts the meaningfully wider formative flows of the beheaded valley across the divide. This difference results in a step change in the unit stream power between the reversed and beheaded channels, potentially leading to a "width-feedback" that promotes ongoing divide migration and reorganization.

Our findings demonstrate that valley width-area scaling is a potential tool for identifying landscapes influenced by drainage reorganization. Accounting for reorganization-specific scaling can improve estimations of erosion rate distributions in reorganized landscapes.

**1 Introduction**

The width of channels and their hosting valleys control river dynamics and functionality with far-reaching implications across a wide range of disciplines, from flood hazards (e.g., Lóczy et al., 2009; Mashael Al, 2010; Sampson et al., 2015), to river ecosystems and habitats (e.g., Beeson et al., 2018; Brussock et al., 1985; May et al., 2013; Sweeney et al., 2004), and hydrological modeling (e.g., Looper et al., 2012). Valley and channel width further play a central role in landscape evolution (Amos and Burbank, 2007; Fisher et al., 2013; Hancock and Anderson, 2002). The relation between valley width, which subsumes channels, terraces, and floodplains, and other measures of valley morphology, including depth and fill thickness, are used to elucidate drainage evolution over geological timescales (e.g., Gibling, 2006; Schumm and Ethridge, 1994) and for inferring past climate changes (e.g., Dury, 1964; Hancock and Anderson, 2002; Marcotte et al., 2021) and tectonic variations (Giaconia et al., 2012). The channel width is a key component in landscape evolution for its control on the shear stress exerted by the flowing water, sediment transport capacity, and erosion rate (Whittaker et al., 2007b; Yanites et al., 2010). Particularly, many landscape evolution and hydrological models approximate the local erosion rate as a function of the channel stream power per unit channel width (Harbor, 1998; Magilligan et al., 2015).

The central role of valley and channel width across disciplines highlights the value of high-resolution width measurements, which could vary by several orders of magnitude within a single basin and across basins and landscapes (Schumm and Ethridge, 1994). Producing high-resolution field-based width measurement of channels and valleys is challenging and time-consuming, and in recent years a growing body of work focused on developing tools for automatic width extraction, based on remotely-sensed data (e.g., Clubb et al., 2022; Fisher et al., 2013; Gilbert et al., 2016; Hilley et al., 2020; Monegaglia et al., 2018; Roux et al., 2014; Rowland et al., 2016). Although these tools represent a significant advancement in river research and management, they commonly focus on specific types of river morphology, and require parameter calibrations, as well as human supervision (Fryirs et al., 2019; Golly and Turowski, 2017). Due to these limitations, in many cases, width of natural channels and valleys is estimated based on the widely recognized scaling relationships between valley and channel widths and fundamental basin properties such as discharge (or its proxy, drainage area), which could be relatively easily measured from digital elevation models (e.g., Lavé and Avouac, 2001; Wobus et al., 2006). Furthermore, channel width - drainage area scaling relationships are frequently used in landscape evolution models, where channel width is implicitly parametrized based on the drainage

area (e.g., Goren et al., 2014; Lague et al., 2014; Shobe et al., 2017; Yanites et al., 2013). However, studies that explored the channel width - drainage area scaling found that it is valid mostly under steady-state conditions, but is less reliable when lithologic, climatic and tectonic complexities are present in the landscape (Allen et al., 2013; Montgomery, 2004; Snyder and Kammer, 2008; Whipple et al., 2013; Yanites, 2018). Consequently, in such landscapes, a more complex scaling involving channel width, area, and slope, was shown to be more applicable (Finnegan et al., 2005). While the influence of tectonic, climatic, and lithologic changes on valley and channel width has been extensively explored (e.g., Allen et al., 2013; Keen-Zebert et al., 2017; Marcotte et al., 2021), the effects of drainage reorganization, which imposes drainage area transiency, were mostly overlooked. The current study targets these effects by exploring valley and channel width scaling under transient conditions that emerge from processes of drainage reorganization.

## 1.1 Width- area scaling in channels and valleys

The common approach for channel width estimation relies on the seminal work of Leopold and Maddock (1953), who used empirical data to establish a power law relation between the channel width, $W$ [m], and discharge, $Q$ [m$^3$/sec]. Combined with the documented correlation between discharge and drainage area, $A$ [km$^2$] (e.g., Dunne and Leopold, 1978), the scaling between channel width and drainage area is often expressed as

$$W = k_c A^d \tag{1}$$

Leopold and Maddock's relation (Eq. (1)) was established for alluvial rivers, where $d$ was found to be ~0.5. A similar scaling was later reported for bedrock rivers, with an exponent that typically ranges between 0.3–0.6 (Kirby and Ouimet, 2011; Montgomery and Gran, 2001; Snyder et al., 2003; Tomkin et al., 2003; Whitbread et al., 2015; Yanites et al., 2010). The exponent's range was mostly attributed to differences in channel bank properties, where more erodible and/or fractured banks widen faster than resistant and intact banks (Spotila et al., 2015; Whitbread et al., 2015; Wohl and Achyuthan, 2002; Wohl and David, 2008). Other studies invoked climatic variations and anthropogenic disturbances to explain variations in the $d$ exponent (Bertrand and Liébault, 2019; Faustini et al., 2009; Snyder et al., 2003).

Although Eq. (1) is commonly used as an empirical relation, it is consistent with process-based theory. Channel widening is attributed to lateral bank erosion induced by particles impacting the channel wall (Li et al., 2020; Turowski, 2018) and is governed by the mechanical properties of the bedload and the channel banks, the channels'

geometry, and the volume and trajectory of the bedload particles (e.g., Finnegan and Balco, 2013; Li et al., 2020; Yanites, 2018). Considering these controlling parameters, Turowski (2018) developed a model relating bedrock channel width to sediment supply, vertical erosion rate, and bank properties. Under spatially uniform erosion rate and steady-state conditions, Turowski's model predicts that the channel width is a power law function of the drainage area, consistent with the form of Eq. (1).

Valley widening occurs when the channel migrates and abuts the valley wall, enabling particles from the channel to erode the valley wall. The effectiveness of valley widening is thus controlled by the frequency at which the channel abuts and erodes the valley wall, which depends on the valley width, channel width, and channel mobility within the valley (which increases with sediment flux) (Clubb et al., 2022). Despite the different processes that underlie the widening of channels and valleys, empirical observations suggest that the relation between the valley width and drainage area follows a similar power law scaling to Eq. (1) (Beeson et al., 2018; Brocard and van der Beek, 2006; Clubb et al., 2022; Langston and Temme, 2019; Langston and Tucker, 2018; May et al., 2013; Schanz and Montgomery, 2016; Snyder et al., 2003; Tomkin et al., 2003). However, the reported range of the exponent $d$ is significantly wider in valleys, ranging between negative values of -0.13 (Clubb et al., 2022) and positive values as high as 1.18 (Beeson et al., 2018). Here too, the exponent range was attributed to differences in the properties of valley-bounding rocks (Brocard and van der Beek, 2006; Keen-Zebert et al., 2017; Langston and Temme, 2019; Schanz and Montgomery, 2016), or, in some high relief landscapes, to the spatial distribution of deep-seated landslides that can cause local recession of the valley walls and, at times, dam the valley and cause upstream aggradation and widening (Beeson et al., 2018; Clubb et al., 2022; May et al., 2013).

**1.2 Width- area-slope scaling relation in channels and valleys**

While the applicability of the simple power low scaling between channel width and drainage area (Eq. (1)) was demonstrated in many settings (Montgomery and Gran, 2001; Whipple et al., 2013; Whitbread et al., 2015; Wohl and David, 2008), field observations show that it is not applicable across all landscapes. Notably, the scaling was demonstrated to fail along areas of localized gradient in rock uplift, e.g., due to local faulting or folding (Allen et al., 2013; Amos and Burbank, 2007; Kirby and Ouimet, 2011; Lavé and Avouac, 2001; Yanites et al., 2010), channels with alternating lithologies (Montgomery, 2004; Spotila et al., 2015), and in channels with transient morphologies due

to temporal changes in rock uplift rate (Whittaker et al., 2007a, 2007b; Yanites, 2018). Finnegan et al. (2005) developed a model for the case of a channel that crosses terrains with variable rock uplift rates. Adopting Manning's equation (Manning et al., 1890) and assuming a constant bankfull width-to-depth ratio along the channel, Finnegan's model predicted that the channel width depends on both the drainage area, as in Eq. (1), and the channel slope, $S$

[m/m]:

$$W = k_b A^b S^c, \tag{2}$$

The exponents $b$ and $c$ in Finnegan's model were calculated to be 0.38 and -0.19, respectively, values that were later supported by observations in various field studies (Finnegan et al., 2005; Kirby and Ouimet, 2011; Spotila et al., 2015; Wright et al., 2022). In studies of transient channel adjustment to changing tectonic forcing, Whittaker et al. (2007a)

and Attal et al. (2008) found that a greater absolute value of -0.44 for $c$ produces better fits for their field observations. The slope dependency in Eq. (2) is consistent with the approach of Turowski (2018) in scenarios where transient conditions are considered, such that the ratio of sediment flux to channel vertical erosion becomes slope dependent.

The significance of including channel slope as a controlling parameter in Eq. (2) depends on the covariance between slope and drainage area. In steady-state drainage networks with uniform lithology, climate, and uplift rates,

the channel slope, $S$, and the drainage area, $A$, covary through a power law relation $S \propto A^{-\theta}$ (Flint, 1974). Therefore, in these cases, the slope can be substituted by the drainage area, and Eq. (2) reduces to the form of Eq. (1). In contrast, the cases where Eq. (2) was found to be a better predictor for channel width (Finnegan et al., 2005; Kirby and Ouimet, 2011; Spotila et al., 2015; Whittaker et al., 2007a; Wright et al., 2022) are those where $S$ and $A$ do not covary, e.g., due to temporal or spatial variations in the environmental conditions.

A theory that relates valley width to drainage area and channel slope in the form of Eq. (2), was provided by Brocard and van der Beek (2006) in settings with alternating alluvial and bedrock sections. In their conceptual model, the inclusion of the channel slope, $S$, as a controlling parameter on the valley width emerges from spatial and temporal variations in the environmental conditions. For example, the channel steepness can serve as a proxy for lithological variations that set the mode of valley widening at different reaches. Alternatively, in scenarios of a channel incising

into a wide flat valley, increased channel slope is often associated with bank steepening, resulting in bank slumping that forms a narrower valley within the preexisting valley bottom. Despite this appealing reasoning, to the best of our knowledge, so far Eq. (2) has not been used to predict valley width in any particular field setting.

### 1.3 Drainage reorganization and width scaling of valleys and channels

Drainage reorganization is widely recognized as an important process affecting the evolution of fluvial systems (e.g., Bishop, 1995; Fan et al., 2018; Harel et al., 2019; Prince et al., 2011; Willett et al., 2014). Reorganization occurs when drainage divides shift through time (Bishop, 1995; Davis, 1889), change basin geometry, and consequently induce changes in the distribution of discharge and drainage area along channels (e.g., Menier et al., 2017; Pechlivanidou et al., 2019). Referring to the width-area scaling in Eq. (1), addition or reduction of the drainage area is expected to result

in channel and valley widening or narrowing, respectively. However, while drainage reorganization is capable of inducing relatively rapid drainage area changes, i.e., following river capture or repeated stochastic events (Shelef and Goren, 2021), width adjustment of channels and valleys most likely requires longer timescales (Brocard and van der Beek, 2006; Wright et al., 2022). Studies that measured channel widths in drainages that experienced recent anthropogenic drainage area perturbations reported ongoing width variations that prevailed for several decades (e.g.,

Jones, 2018; Snyder and Kammer, 2008). Based on a theoretical model, Turowski (2020) postulated that the timescale of channel width adjustment to discharge perturbations is in the order of thousands of years. For valleys, the time-gap between the change in drainage area and width adjustment is expected to be even longer, most likely in the order of tens of thousands of years (Hancock and Anderson, 2002; Langston and Tucker, 2018), because valley width represents the channel location integrated over long periods (Schumm and Ethridge, 1994; Tomkin et al., 2003).

Although the potential scaling deviation following reorganization is highly consequential for fluvial landscape functionality, the effects of reorganization on fluvial channel and valley width scaling have not yet been evaluated. We hypothesize that this scaling, particularly that of the valleys, expresses the delayed response of width adjustment to drainage area changes following reorganization. Accordingly, the coefficient and exponent values that relate drainage area with valley width in reorganized drainages could meaningfully deviate from drainages that did

not experience reorganization.

To test this hypothesis and to evaluate the effect of reorganization on width-area scaling of valleys and channels, we analyzed and compared the geometry of reorganized and undisturbed drainages in the southern Negev desert, Israel, where drainage reorganization is well established by field observations (Avni et al., 2000; Ginat et al., 2000, 2002; Harel et al., 2019). In the current analysis, we aim to (i) explore if and how the scaling between valley

width and drainage area in Eq. (1) varies between reorganized and non-reorganized drainages and among drainages that experienced different modes of reorganization; (ii) In cases when drainage area and slope do not covary, study the independent slope influence on valley width scaling following Eq. (2); (iii) compare the adjustment of channel width relative to valley width following reorganization; and (iv) examine landscape evolution implications of the valley and channel widths scaling in reorganized drainages.


## 2. Study area

### 2.1 Geologic and Geomorphic setting

We explore channel and valley width scaling along ephemeral drainage networks that incise into the southeastern Negev Highlands, Israel (Fig. 1). The highlands are bounded to the east by ~400-600 m high cliffs which rise above

the Arava Valley, part of the transtensional Dead Sea plate boundary with a rift-like structure that stretches between the Dead Sea and the Gulf of Aqaba (Garfunkel, 1981; Garfunkel et al., 2014, Figs. 1a-b).

The main drainage divide in the study area separates steep east-flowing basins that drain across the cliff toward the Arava Valley, from west-flowing, low relief basins that flow on the Negev Highlands (Fig. 1b-d). The lithology exposed along the highland valleys consists primarily of Cretaceous limestone and dolomite strata (Ginat,

1991). The climate is hyper-arid with an average annual precipitation of ~<50 mm (Bitan and Rubin, 1991), typically generating one to few flashflood events per year. These climatic conditions generally persisted through most of the Pleistocene (Amit et al., 2006, 2011), except for short episodes of wetter conditions (Ginat et al., 2018; Vaks et al., 2013).

The eastern Negev desert has been experiencing ongoing fluvial reorganization since the late Miocene (Avni

et al., 2000, 2012). Before the development of the Arava Valley, rivers that originated in the Jordanian highlands, east of the Arava Valley, flowed westward, crossing the plate boundary along the Negev Highlands towards the Mediterranean (Garfunkel and Horowitz, 1966; Zilberman and Calvo, 2013). Since the Miocene, tectonic activity along the Dead Sea plate boundary formed the Arava Valley, which gradually became a prominent base level. Consequently, during the Plio-Pleistocene, several large-scale capture events redirected major drainage systems in the

Negev toward the central Arava valley (Avni et al., 2000; Ginat et al., 2000, 2002; Guralnik et al., 2010). Field

observations and a regional χ analysis (a morphometric parameter used to approximate the stability of drainage divides, (Willett et al., 2014)) suggest that the regional divide between the Arava Valley and the Negev Highlands is still actively migrating westward (Harel et al., 2019).

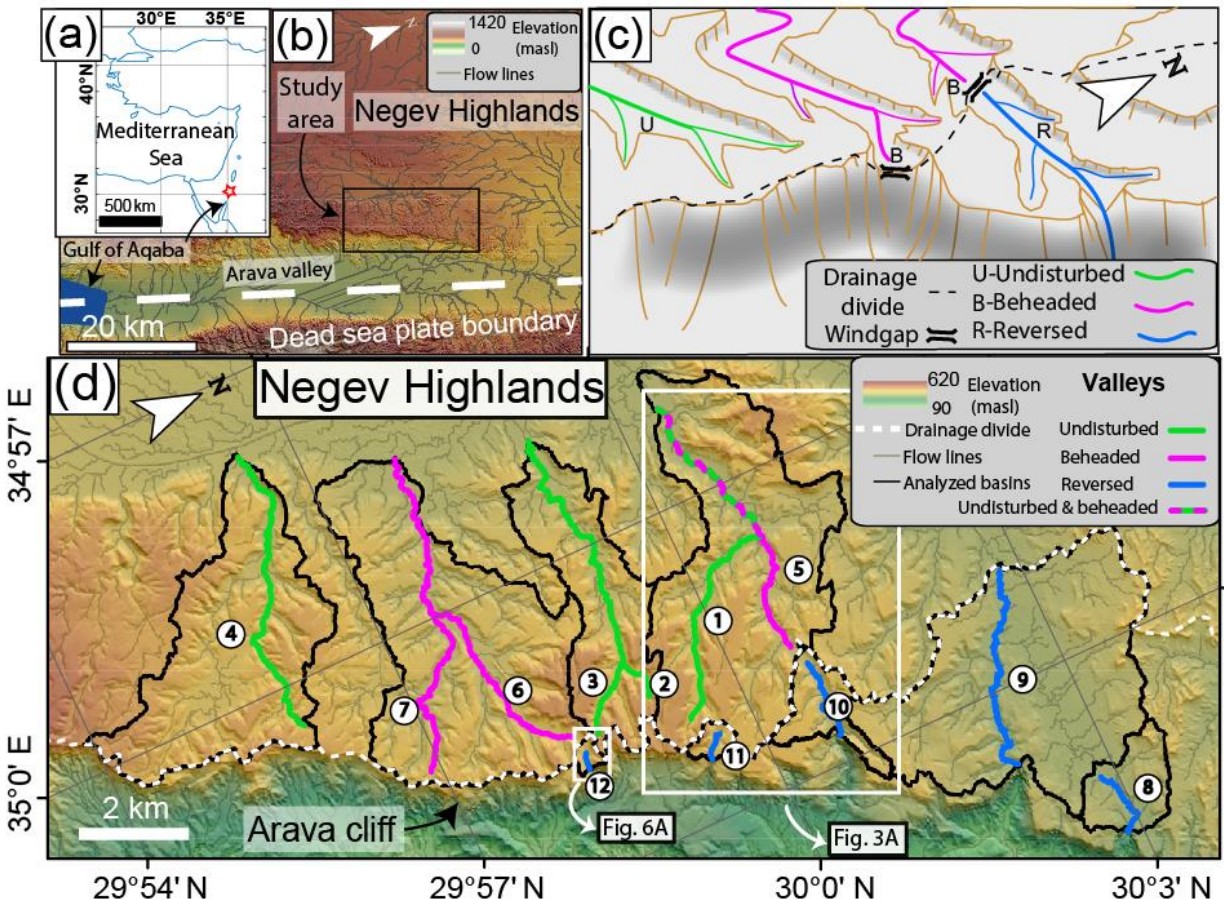

**Figure 1 (a)** Orientation map with coastlines (blue) showing the study area location (red star). **(b)** Shaded elevation map, illustrating the regional rift-morphology along the plate boundary (dashed white line) adjacent to the study area (black rectangle). The maps in (b) and (d) are based on TanDEM-X 0.4 arcsec DEM (Wessel, 2016). **(c)** A simplified sketch of valley categorization in the study area: Undisturbed valleys (Green, 'U' tag) are valleys that do not intersect with the cliff and are minimally affected by drainage reorganization. Beheaded valleys are valleys that were beheaded due to cliff retreat or drainage reversal (pink, ''B' tag), and Reversed valleys (blue, ''R' tag), that presently flow toward the cliff, are commonly recognized by their barbed tributaries which join the main channel at a >90 degrees angle. **(d)** A shaded elevation map of the study area, illustrating the drainage divide (white dashed line) between the Negev Highlands and the Arava Valley. The basin boundaries (black lines) are defined by the valley section's outlet. Encircled numbers refer to the valley serial numbers in Tables 1 and 2.

**2.2 Categories and characteristics of valleys in the study area**

To explore the effects of drainage reorganization on the valley width scaling relations, we analyzed 12 valley sections associated with different drainage reorganization categories. All sections are located adjacent to the Negev-Arava drainage divide (Fig. 1d), resulting in relatively small drainage areas of 0.2-14.2 km$^2$. The valleys are incised into bedrock, generating relief of several tens of meters between the valley bottoms and the highlands' flat interfluves. The valleys were classified into three categories based on the association of the valley to the Arava cliff (Fig. 1c-d), the morphology of the valley section (Fig. 2), and additional supporting field observations. The three categories are:

1. **Undisturbed valleys (n=4)** are westward flowing valley sections whose headwaters are adjacent to the cliff line and are not meaningfully beheaded. In some cases, field evidence indicates that some portions of the drainage area along the low-relief interfluves were lost due to divide migration associated with cliff receding. Yet, the receding cliff does not intersect the incised portion of the valleys; therefore, these valleys are referred to as 'undisturbed.' In these valleys, the low-order (sensu Strahler) incised segments are characterized by a V-shaped morphology (Figs. 2a and 2c) with a bedrock valley bottom that is several meters wide. Farther downstream, typically at a distance less than 1 km from the valley head, the valley bed becomes alluviated, and its width increases to tens of meters. Higher-order valleys widen downstream at slower rates than low-order valleys and typically have a trapezoid cross-section with a sediment-filled flat valley bottom and steep valley walls (Figs. 2b-d). In the undisturbed and beheaded categories (below), the entire valley is typically occupied by a low-relief, braided and dynamic channel system. Field observations of fully flooded valley bottom during large rainstorm events (Fig. 2d), suggest that the formative flow width is the entire width of the valley bottom.

2. **Beheaded valleys (n=3)** are west-flowing sections whose headwaters were beheaded. Beheading is indicated by a windgap, i.e., a flat, valley-confined drainage divide located along the cliff or shared with a reversed valley (described below), indicating the truncation of an incised paleo-valley that likely drained a larger area. Close to the windgap, the beheaded valleys are characterized by a U-shaped cross-section (e.g., Figs 1c, 2e-f, 3a, and 6a), likely controlled by the concave profiles of side colluvial aprons. West and downstream from the windgap, beheaded valleys become indistinguishable from the undisturbed valleys with the trapezoid-shaped cross-section. Valley's beheading is associated with either the receding cliff and its coinciding divide, or with localized divide migration within the valley as part of a reversal process on the opposing side of the windgap (e.g., Bishop, 1995; Harel et al.,

2019, Shelef and Goren, 2021). The beheaded valleys bear a similar valley bed morphology to the undisturbed

category; thus, the formative flow width of the beheaded valleys is the entire width of the valley bottom.

    3. **Reversed valleys (n=5)** host east-flowing channels that reversed their drainage direction (Bishop, 1995) from west to east. These valleys are bounded between a windgap on the west side and a knickpoint on the east, where the channel flows across the cliff (e.g., Figs 2e, 3a and 6a). Harel et al. (2019) identified these sections as reversed drainages based on the presence of barbed tributaries and west grading terraces that record the antecedent valley

gradient, which is opposite to the present-day channel's drainage direction. The reversed valley sections share windgaps with beheaded valleys, indicating that they were part of an antecedent west-flowing drainage (Harel et al., 2019). The two northern reversed valleys (8 and 9 in Fig. 1d), initiate in a E-W trending strike valley which dictates a wide windgap (>500m, Ginat, 1997), whereas downstream from the divide they exhibit trapezoid cross-sections. In the three other valleys, the windgap is U-shaped, and downstream the channel incises into alluvial-colluvial valley

fill, creating cut terraces and forming a V or box-shaped channel cross-section within the broader valley (e.g., transect d-d' in Figs. 2e-f). In most cases, close to the knickpoint where the channel crosses the cliff, it incises into bedrock, and the valley cross-section changes to a V-shaped morphology (e.g., transect c-c' in Figs. 2e-f).

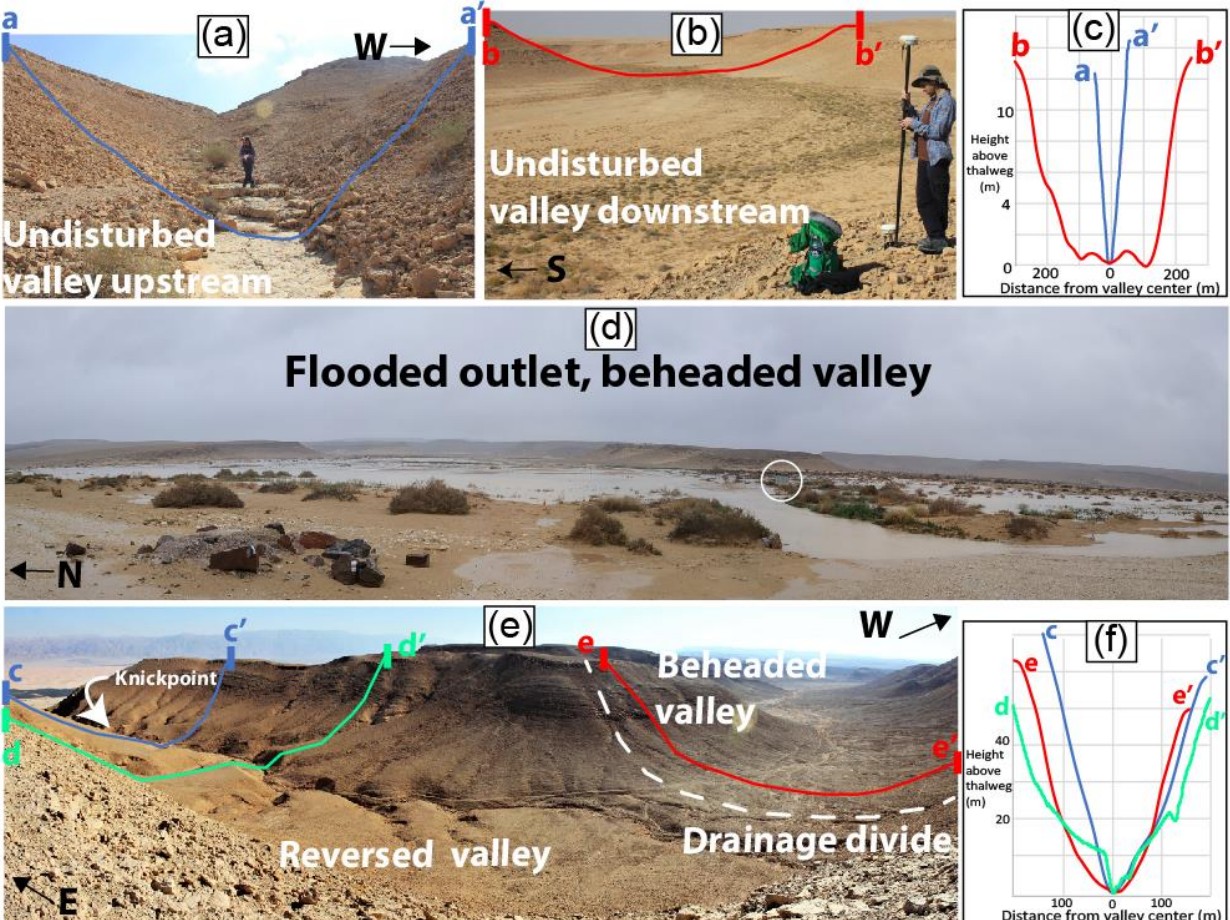

Figure 2: Field photos and valley transects of valley sections in the study area. (a,b) Upstream and downstream segments of undisturbed valleys ((a) and (b), respectively). The drainage area in panel (a) is 0.08 km², and in panel (b) is 1.85 km². Blue and red lines (a-a' and b-b,' respectively) mark the cross-section profiles shown in panel (c). (c) Transects of a-a' and b-b.' Note the V-shaped transect near the valley head relative to the trapezoid morphology of the downstream section. (d) Flooded valley bottom at the outlet of two beheaded valleys, 6 and 7, after an intense rain event in February 2020. An ~1.5 meter wide sign is encircled for scale. (e) Panorama of reversed and beheaded valleys (valley 12 and 6 in Table 1 and Fig. 1d), and the confined, flat windgap between them. The c-c' transect (blue) was measured near the knickpoint at the edge of the reversed section, d-d' (green) follows the terraces representing the paleo-valley and the channel that incises into them, and e-e' (red) was measured close to the windgap on the beheaded side. (f) Cross-sections of transects c-c', d-d,' and e-e,' emphasizing the difference between the U-shape transect near the windgap (e-e'), the V-shaped channel profile incised into the U-shaped valley terraces (d-d'), and the V-shaped valley transect above the knickpoint (c-c').

**3. Methods**

We studied the effect of reorganization on the width scaling of valleys by exploring the coefficients and exponents that control valley width variation, following equations (1) and (2). Valley width - drainage area scaling, based on Eq. (1), is explored for all valley sections in our study area, and the role of the slope is explored through Eq. (2) only for the reversed sections that generally show poor correlations between slope and drainage area. In one reversed section, we focus on the scaling between channel width, drainage area and channel slope that emerge through Eq. (2).


**3.1 Drainage area and slope extraction**

Elevation data were derived from TanDEM-X (Wessel, 2016) with 0.4 arcsec resolution (~11.6 m/pixel in the field area). The drainage area was extracted from a flow accumulation raster, computed using a D8 flow routing algorithm (O'Callaghan and Mark, 1984). The threshold drainage areas used for defining the flow network are specified in Table

S1 in the Supplement. The channel slope used for exploring slope-area relations and channel and valley width predictions following Eq. (2) was estimated along the flow network (thalweg) by using the slope of a linear regression between elevation and distance over a centered 7-pixel running window.

**3.2 Valley width measurements**

To compute the coefficients and exponents of Eq. (1) and (2) that best fit the geometry of valley sections in the study area, we extracted the valley widths along the analyzed valley sections. In the undisturbed and beheaded categories, the valley width refers to the flat valley bottom that is fully flooded during formative floods, while in the reversed category the valley width typically includes terraces that preserve former levels of the valley bottom (Harel et al., 2019). Unlike the upstream drainage area and slope, which are derived through relatively simple calculations over the

DEM, defining and extracting the valley width based on a DEM is not straightforward and requires a tailored procedure (Clubb et al., 2017, 2022; Golly and Turowski, 2017; Hilley et al., 2020; Roux et al., 2014; Rowland et al., 2016; Sechu et al., 2021). Particularly, the location and orientation of valley width measurements require caution because the width is often not well-defined in proximity to side tributaries and valley bends (Beeson et al., 2018; Clubb et al.,

2022). To overcome these challenges, we developed a semi-automatic approach for optimal measurements of valley

width.

Valley width was measured by applying two consecutive operations. First, a polygon representing the valley bottom is extracted, and second, valley width is measured over the valley bottom polygon at optimal points (Fig. 3a). The first step is achieved by applying the ArcGIS plugin VBET- 'Valley bottom extractor tool' (Gilbert et al., 2016). VBET identifies valley boundaries based on user-defined slope thresholds, representing the transition from the valley

bottom to the hillslope. This method particularly suites valley morphologies where the valley bottom can be easily distinguished from the valley walls based on a distinct slope break which is the case in most of the studied valley sections. VBET parameters used for the current analysis are described in Table S1 in the Supplement. Importantly, these parameters were fitted to each basin (each including one or two sections) separately, by an iterative process of visually comparing the valley bottom polygons against 0.5m/pixel aerial orthophotos and fine-tuning the parameters

to achieve the best visual fit. In six basins, this procedure was not sufficient to achieve a satisfying fit between the VBET polygon and the orthophoto, mainly due to local DEM inaccuracies. In these cases, the polygons were manually edited to correct local mismatches, based on the orthophotos, available topographic data, and field observations. In five of the edited polygons, the area difference between the original and the edited polygons was < 5%; in one case, the area difference was 10% (Table S1 in the Supplement). Shapefiles of the polygons before and after manual editing

are available in the Supplement.

Valley width measurements over the VBET polygons were achieved by applying an ArcGIS-based algorithm that identifies points that are sufficiently far from bends and confluences and are located along the valley centerline. In these optimal locations, valley transects are taken perpendicular to the centerline, whose length represents the valley width. The final output is a set of pixels located at intersections between the thalweg and the valley transects, which

are assigned with valley width, drainage area, and slope values (e.g., Fig 3a). The algorithm is described in detail in Section S1 and Figs. S1-S5 in the Supplement.

### 3.3 Regression analysis

As a preliminary step to explore valley width scaling, the covariance between slope $S$ *[m/m]* and drainage area $A$ *[km]*

was quantified using linear regression over a binned log-transformed values (e.g., Wobus et al., 2006). For all valley

sections, the best-fit values of $k_c$ and $d$ in Eq. (1) were calculated by using a least-squares linear regression over log-transformed $W$ $[m]$ and $A$ $[km^2]$ (e.g., Fig. 3b). We used units of $[km^2]$ for drainage area to facilitate direct comparison with prior studies that conducted a similar analysis using these units (e.g., Clubb et al., 2022; Langston and Temme, 2019; Schanz and Montgomery, 2016; Tomkin et al., 2003). In the reversed category, the slope and area do not always covary, hence, in this category we used multivariate least-squares linear regression over log-transformed $W$ $[m]$, $A$ $[km^2]$, and $S$ $[m/m]$ to find the best-fit values of $k_b$, $b$, and $c$ in Eq. (2) (following Attal et al., 2008; Spotila et al., 2015) (e.g., Fig. 3c). In the regressions used for Eqs. (1) and (2), the data were not binned.

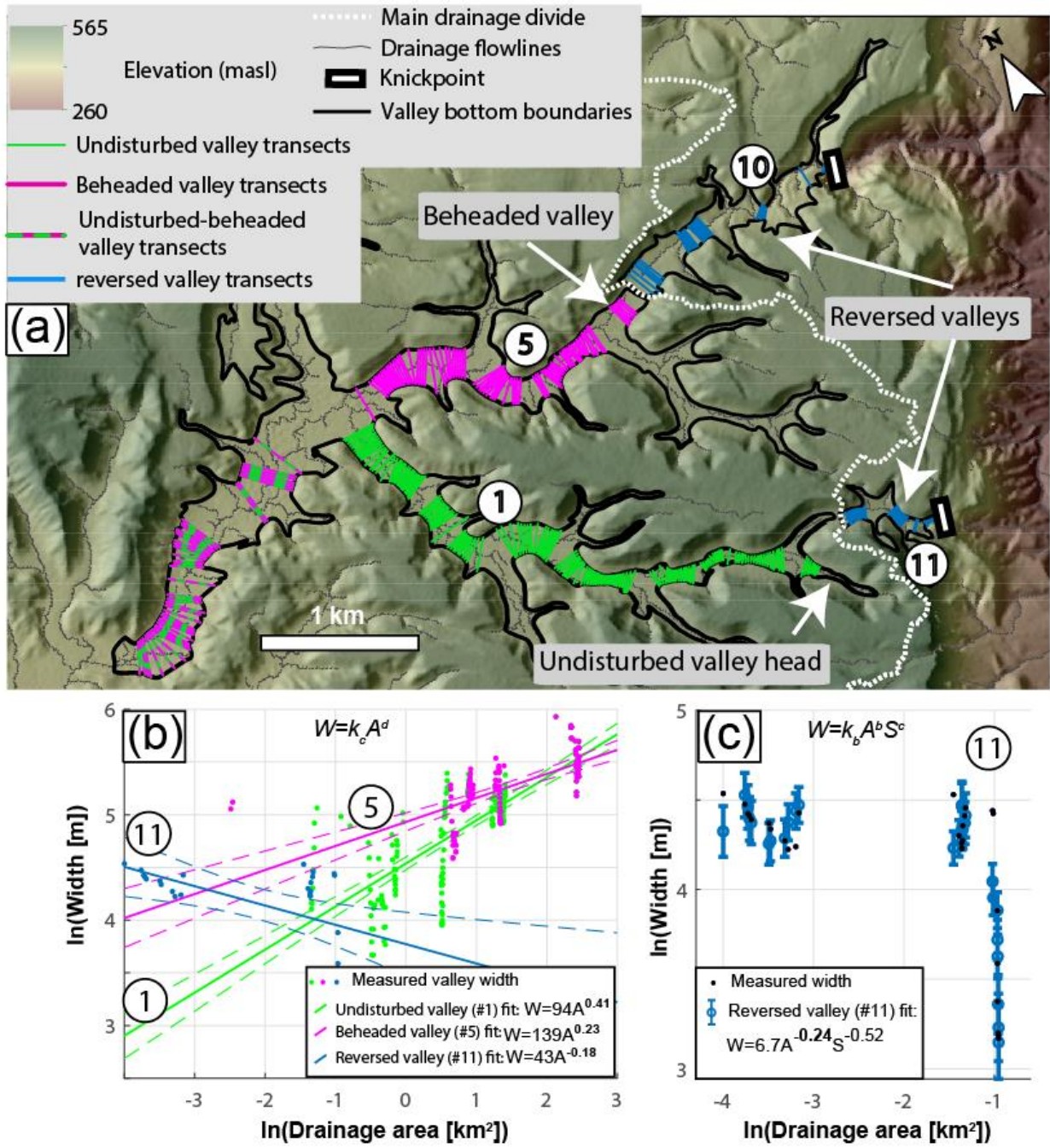

**Figure 3: Valley width measurements and regression-based models for valleys 1, 5, and 11 from Fig. 1d and Tables 1 and 2. (a)** A valley bottom polygon (black) overlies a shaded elevation map based on TanDEM-X 0.4 arcsec DEM (Wessel, 2016). Green, pink, and blue lines represent transects of undisturbed, beheaded, and reversed valley sections, respectively. Dashed lines represent measurements at valley sections downstream of confluences between undisturbed and beheaded valleys. The reversed valleys extend between the main drainage divide (dashed white curve) and knickpoints (white boxes located at the cliff-flowlines intersections). **(b)** Linear regression fitted lines from log-transformed valley width and drainage area, for the undisturbed, beheaded, and reversed valleys 1, 5, and 11, respectively. The dashed lines represent 95% confidence bounds. The equations in the bottom right are the linear models' $k_c$ coefficients and $d$ exponents. **(c)** Multivariate regression results with the associated $k_b$ coefficient and $b$ and $c$ exponents for the reversed valley 11. The 95% confidence interval is represented by error bars.

### 3.4 Detailed analysis of channel and valley width

In contrast to the undisturbed and beheaded categories, in the reversed category, the valley and the channel are decoupled. In this category, we examined how fitting the valley width compared to the channel width affects the predictors $k_b$, $b$, and $c$ in Eq. (2). Valley 12 (Fig. 1d and Tables 1 and 2) is a thoroughly surveyed site (Harel et al., 2019) that was chosen for this analysis. The channel parameters are based on a 15 cm/pixel DEM and a 3 cm/pixel orthophoto generated using a structure from motion (SfM) algorithm over drone-acquired aerial photos (80% overlap). Here, the sub-meter scale topography of the high-resolution DEM inhibited the VBET tool from discriminating the channel bottom precisely, and therefore the channel bottom polygon was delineated manually based on the 15 cm/pixel DEM and orthophoto. Then, the valley width measurement algorithm, described in Sect. 3.2, was applied over the channel polygon. The drainage area and elevation data were extracted from the 15 cm/pixel DEM. The slope was calculated following the procedure described in Sect. 3.1, with a running window of 541 pixels, such that the length of the along-flow distance covered by the window was comparable to that used for the valleys. Finally, the best-fit $k_b$, $b$, and $c$ values were calculated using a multivariate least-square linear regression, as described in Sect. 3.3.

### 3.4 Validations and errors in the measurements and model

The main potential sources of valley width measurement errors originate from the DEM horizontal resolution, $R$ (~11.6 $m^2$/pixel), and the relative vertical accuracy (~2m, Wessel, 2016). To incorporate the uncertainty stemming from the horizontal resolution of the DEM, we assigned each valley width measurement with a constant error, evaluated as $\sqrt{2}R$.

To independently explore the effect of inaccuracies in the DEM on the final valley width measurements, seven valley transects were measured with a differential GPS (DGPS). The valley bottom was extracted from the transects by applying the same criteria used to identify the slope-break that was applied in VBET to that basin. The DEM-based and DGPS-based valley width measurements and their relations are shown in Fig. S6 and in Table S2 in the Supplement. The differences between the DEM-based measurements relative to the DGPS-based measurements range between 2-20 m and are scale-independent. The percent deviation between the measurements is <0.3%, except

for the narrowest valley, where ~3 m difference between the DEM and the DGPS-based measurements yielded a percent deviation of ~25%. Overall, the mean percent deviation is 3.7%, and the RMSE is 13 m (whereas the mean valley width of the seven transects is 110 m), smaller than the resolution-associated error of $\sqrt{2}R$.

## 4. Results

### 4.1. Slope area correlation

The slope-area relation of the studied valley sections is presented in Fig. S7 in the Supplement. The $R^2$ of the slope-area regressions in the undisturbed and beheaded valleys range from 0.68 to 0.93. In the reversed valleys, the $R^2$ of two valley sections is ~0.5, and the $R^2$ of the other three reversed valley sections is $R^2 < 0.14$. As mentioned in Sect. 1.2, when slope and area strongly covary, Eq. (2) reduces to the form of Eq. (1). For that reason, while the valley width – drainage area scaling (Eq. (1)) is computed for all valley categories, Eq. (2) is applied only for the reversed valley sections where the slope-area covariance is low.

### 4.2 Valley width - drainage area scaling

The best-fit coefficients and exponents of the valley sections, their 95% confidence intervals, and the $R^2$ for the regression are presented in Table 1 and Fig. 4. P-values of the predictors and the least-square regressions are provided in Table S3 in the Supplement. The regressions are depicted in Fig. S8 in the Supplement.

Table 1: Regressions for Eq. (1), $W = k_c A^d$, all valley sections.

| Valley category | Valley ID | $k_c$ ($10^{-6}$ m$^{(1\text{-}2d)}$), (Min.-Max 95% confidence interval) | $k_c$ ($10^{-6}$ m$^{1\text{-}2d}$) Median, (Min.-Max.) | Area exponent $d$, (95 % confidence interval) | $d$ Median, (Min.-Max.) | $R^2$ |
|---|---|---|---|---|---|---|
| Undisturbed | 1 | 94, (88-99) | 100, (94-110) | 0.41±0.04 | 0.47, (0.26-0.54) | 0.64 |
| | 2 | 106, (102-110) | | 0.54±0.02 | | 0.93 |
| | 3 | 110, (106-113) | | 0.54±0.02 | | 0.94 |
| | 4 | 67, (63-71) | | 0.26±0.04 | | 0.45 |
| Beheaded | 5 | 139, (127-151) | 139, (123-168) | 0.23±0.05 | 0.18, (0.15-0.23) | 0.42 |
| | 6 | 168, (158-177) | | 0.15±0.04 | | 0.37 |
| | 7 | 123, (120-127) | | 0.18±0.02 | | 0.73 |
| Reversed | 8 | 131, (95-182) | 101, (24-1378), | -0.74±0.45 | -0.56, (-1 - (-0.18)) | 0.37 |
| | 9 | 1378, (377-5041) | | -1±0.53 | | 0.23 |
| | 10 | 101, (90-113) | | -0.24±0.07 | | 0.69 |
| | 11 | 43, (32-59) | | -0.18±0.13 | | 0.26 |
| | 12 | 24, (12-49) | | -0.56±0.3 | | 0.64 |


The least-square regression results reveal unique ranges of the drainage area exponents, $d$, for each predefined valley category (Fig. 4b). The undisturbed valleys are characterized by the highest exponents, ranging from 0.26 to 0.54, whereas the $d$ exponents of the beheaded valleys are lower, 0.15- 0.23. Uniquely, the reversed valleys have negative $d$ exponents, ranging from -0.18 to -1, indicating that in this category, the valleys narrow with increasing drainage area.


Unlike the $d$ exponent values, the $k_c$ coefficients are non-unique for the different valley categories (Fig 4a). The values of the $k_c$ coefficient, which represents a valley width at $A=1$ [km$^2$], range from 94 to 110 ($10^{-6}$ m$^{1\text{-}2d}$) in the undisturbed valleys which differs from range of the beheaded valleys category, 123- 168 ($10^{-6}$ m$^{1\text{-}2d}$). The $k_c$ coefficient values for the reversed valleys show a large variability across three orders of magnitude ranging between 24 ($10^{-6}$ m$^{1\text{-}2d}$) and 1378 ($10^{-6}$ m$^{1\text{-}2d}$).


The performance of the power law model (Eq. 1) was evaluated through the value of $R^2$. In the undisturbed and beheaded categories, $R^2$ ranged between 0.37-0.94. In the reversed valleys, two valleys show $R^2$ values of 0.64 and 0.69, and the three other valleys exhibited lower values of 0.23-0.37 (Table 1). The $W$-$A$ relations are statistically significant for all valleys ($\alpha$=0.05, Table S3 in the Supplement).

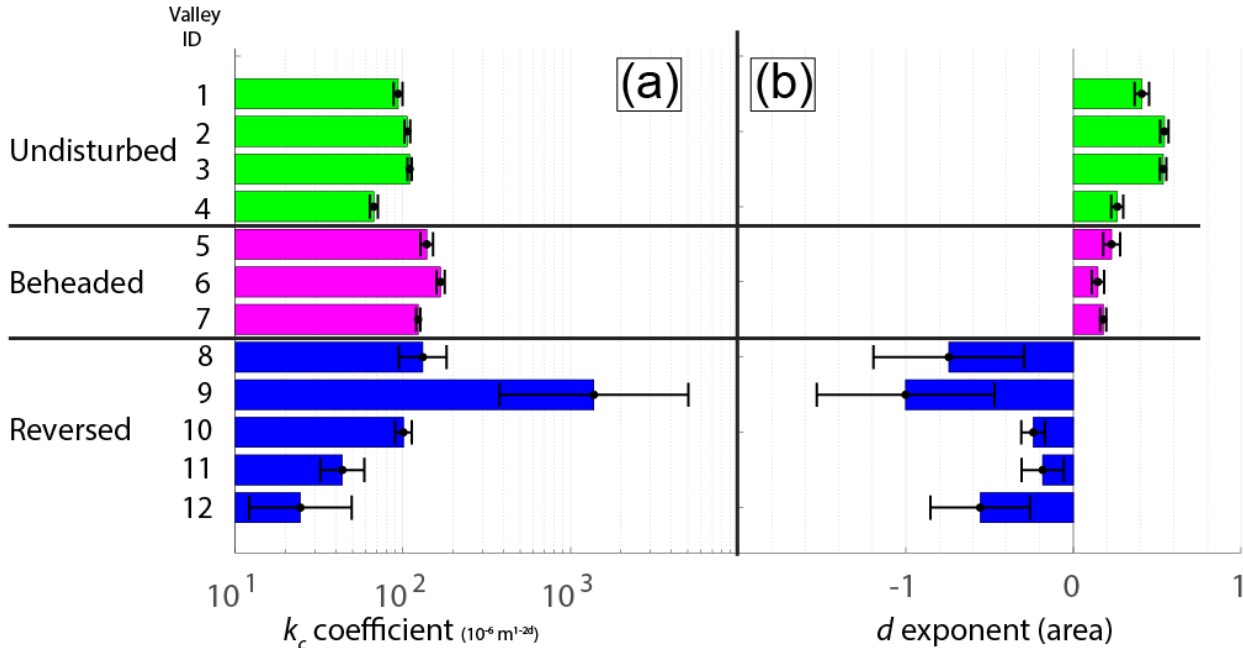

Figure 4: Bar plots of the $k_c$ and $d$ values (panels (a), (b), respectively) for valley sections of the categories defined in Fig. 1c.
The error bars represent the 95% confidence interval. Unlike the $k_c$ values, which lack a clear trend, the values of the $d$ exponent (b) fall within a distinct range for each valley category. Note the log-scale of the x-axis in panel (a).

### 4.3 Valley width - drainage area -slope scaling in the reversed category

The results in Sect. 4.1 demonstrate that most reversed valleys are characterized by a poor correlation between slope and drainage area. Therefore, in this category, Eq. (2) may yield a better prediction for the valley width as a function of both the drainage area and slope. In Table 2 and Fig. 5, we present the results of this multivariate regression, including 95% confidence intervals and adjusted $R^2$. P-values of the predictors and of the multivariate regressions are provided in Table S4 in the Supplement.


Table 2: Regressions for Eq. (2), $= k_b A^b S^c$ for reversed valley sections.

| Valley category | Valley ID | $k_b$ ($10^{-6}\ m^{(1-2b)}$), (Min.-Max 95% confidence interval) | $k_b$ ($10^{-6}\ m^{1-2b}$) Median, (Min.-Max.) | Area exponent $b$, (95 % confidence interval) | $b$ Median, (Min.-Max.) | Slope exponent $c$, (95 % confidence interval) | $c$ Median, (Min.-Max.) | Adjusted $R^2$ [a] |
|---|---|---|---|---|---|---|---|---|
| Reversed | 8 | 14, (9-23) | 14, (2-561) | -0.32±0.21 | -0.32, ((-0.98)-(-0.23)), | -0.66±0.14 | -0.51, ((-0.91)-0.05) | 0.89 |
| | 9 | 561, (146-2151) | | -0.98±0.49 | | -0.16±0.11 | | 0.33 |
| | 10 | 125, (100-155) | | -0.23±0.06 | | 0.05±0.04 | | 0.74 |
| | 11 | 7, (4-10) | | -0.24±0.06 | | -0.52±0.12 | | 0.82 |
| | 12 | 2, (1-6) [b] | | -0.43±0.16 | | -0.91±0.37 | | 0.92 |


[a] Adjusted $R^2$ is used here for conservativeness

[b] Predictor P-value >0.05. See Table S4 in the Supplement.

The results of the multivariate regression based on Eq. (2) demonstrate that in the reversed valley sections, the drainage area exponent, $b$, remains negative and is within the range of -0.98 to -0.23, similar to the drainage area exponent, $d$ computed based on Eq. (1) (Fig. 5b). The $k_b$ coefficients are between 2 and 561 ($10^{-6}\ m^{1-2b}$) (Fig. 5a). The values of the slope exponent, $c$, are negative, between -0.91 and -0.16, except for valley 10, where the exponent is about zero (Fig. 5c). With the exception of valley 9, the adjusted $R^2$ of the model is between 0.74-0.92. Overall, all the adjusted $R^2$ values based on Eq. (2) are higher than the standard $R^2$ obtained based on Eq. (1).


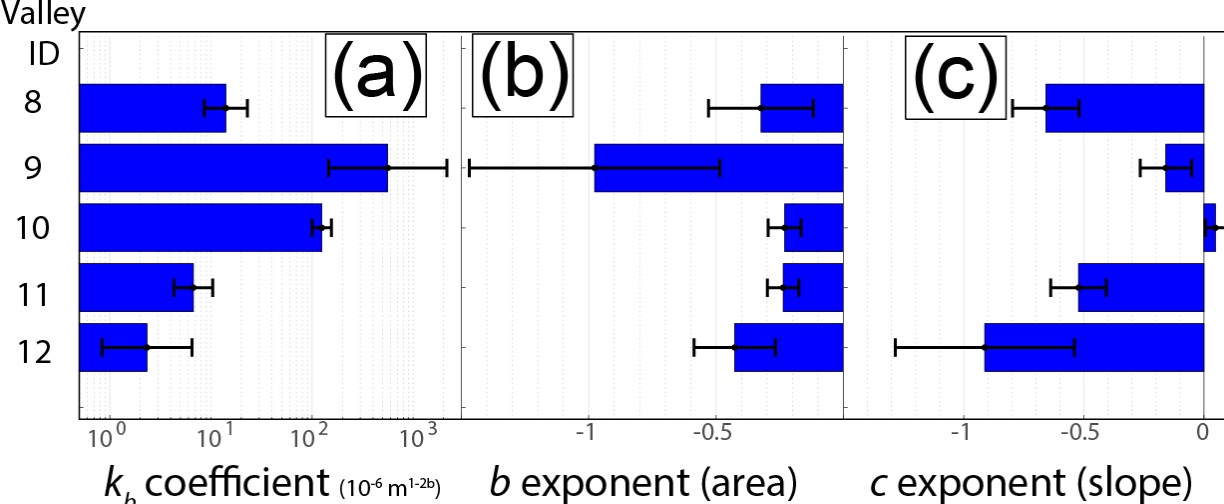

**Figure 5: Bar plots of the $k_b$ coefficient and $b$ and $c$ exponents in Eq. (2) for the reversed valley category (panels (a), (b), and (c) respectively), fitted by multivariate regression. Error bars represent the 95% confidence interval. The $k_b$ values show a large variability. The area exponents, $b$, are generally less negative than the $d$ exponents fitted to Eq. (1) in Table (1) and Fig. (4). Except for valley 10, the slope exponents $c$ have negative values. In valleys 8, 11, and 12, the $c$ exponent is more negative than the area exponent, $b$, reflecting the key role of slope in reversed valley width prediction. Note the log-scale of the x-axis in panel (a).**

## 4.4 Comparing valley width and channel width in a reversed drainage

To explore the effect of the drainage area and slope on the width of the channel vs. the width of the valley in a reversed valley section, where the valley and the channel are decoupled, we extracted the predictors $k_b$, $b$, and $c$ in Eq. (2) for channel width in the reversed valley 12 (Table 2, Fig. 6). The channel in valley 12 initiates east of the windgap and incises into the erodible valley fill, where it merges with short side tributaries. Farther downstream, it merges with a barbed tributary that joins the valley from the north (Fig. 6a). At the barbed tributary junction point, the reversed channel is incised ~15 m below the surface of the antecedent valley bottom. Approximately 160 m farther downstream, bedrock is exposed at the base and the north bank of the channel. The channel traverses the escarpment 40 m downstream where it forms a steep knickpoint that marks the edge of the reversed section.

Field observations show that while the reversed valley narrows downstream (i.e., eastward), the channel width increases in this direction (Fig. 6a), a pattern that was observed also in other reversed valley sections. In valley 12, a multivariate regression over the channel data reveals a drastically different dependency between the channel's width, drainage area, and slope compared to the valley (Fig 6b). For the channel, the least-square multivariate regression ($R^2$=0.72) yields a $k_b$ coefficient of 8 *($10^{-6} m^{1-2b}$*, with a 95 % coefficient interval of 3-17), a positive and high

*b* exponent of 0.62±0.18, and a negative, statistically insignificant *c* exponent of -0.24±0.25. In contrast, the computed

values for the valley are 2 *(10⁻⁶ m¹⁻²ᵇ*, with a 95 % coefficient interval of 1-6*)* for $k_b$, a negative *b* exponent of -0.43±0.16,

and a statistically significant *c*, -0.91±0.37.

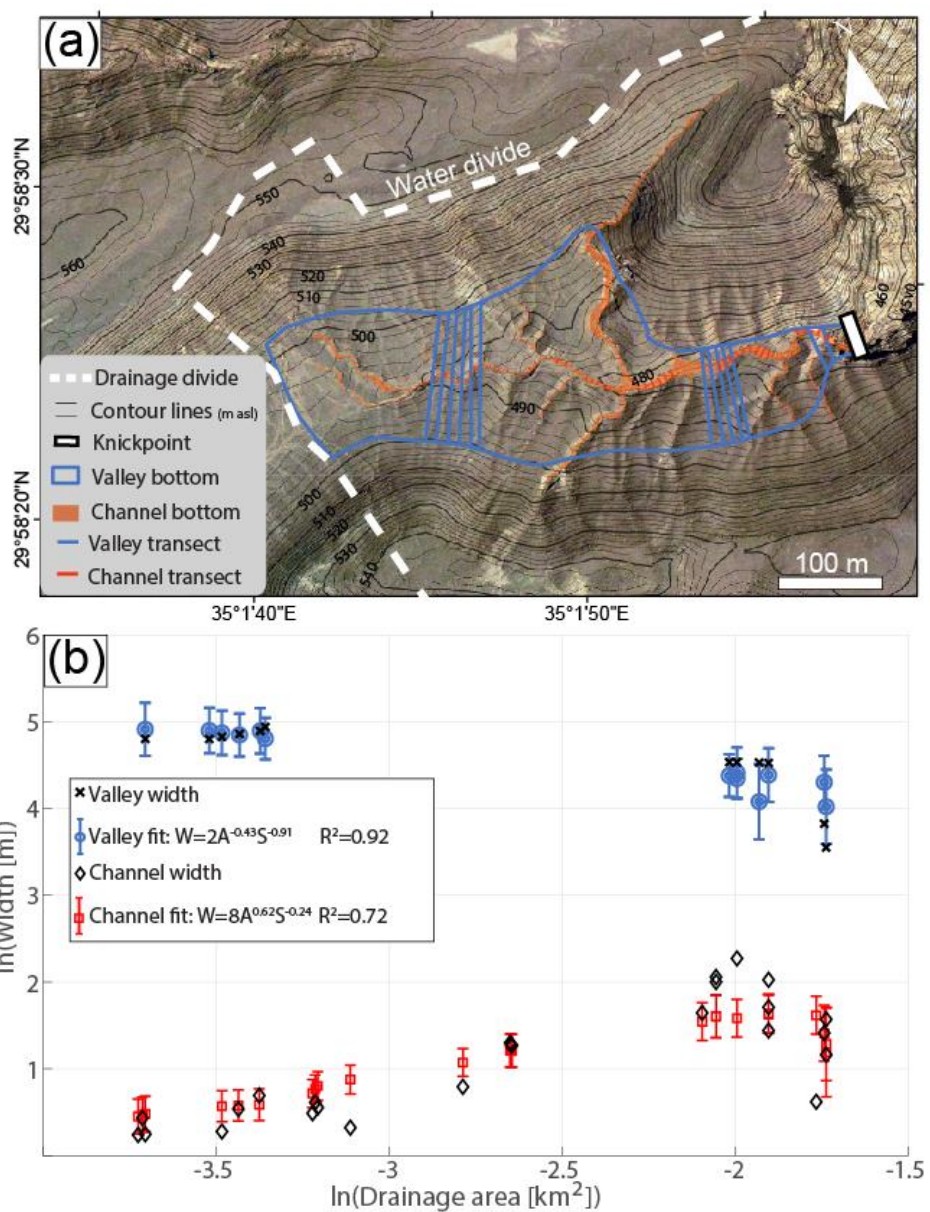

**Fig 6: Variations in valley and channel widths along a reversed section (valley 12 in Tables 1 and 2, and Fig. 1d). (a) Valley**
**(blue) and channel (orange) polygons and width transects, sketched over a 0.5 m resolution orthophoto. The slope break**
**between the valley bottom and the hillslope is emphasized by the density change of the 2m contour lines (thin black lines).**
**(b) Width-area-slope scaling of reversed valley width (data in black, fit in blue), which narrows with drainage area,**
**contrasting with the width of the channel (data in black, fit in red) that increases downstream. This difference is expressed**
**by the drainage area exponent, *b*, which is positive for the channel and negative for the valley.**

## 5. Discussion

### 5.1. Drainage reorganization affects the scaling of valley width – drainage area

The width-area regression results reveal that undisturbed, beheaded, and reversed valleys are characterized by a distinct range of the drainage area exponent, *d,* indicating the fingerprint of reorganization in the width-area scaling

of valleys. In our study area, the *d* exponent values of the undisturbed valleys range between 0.26 and 0.54, consistent with previously published exponent values for valleys (Beeson et al., 2018; Brocard and van der Beek, 2006; Clubb et al., 2022; Langston and Temme, 2019; Schanz and Montgomery, 2016; Shepherd et al., 2013; Snyder et al., 2003; Tomkin et al., 2003). The beheaded valleys in the study area are characterized by *d* exponent values of 0.15-0.23. While this range partly overlaps with previously published valley width-area scaling (Beeson et al., 2018; Clubb et

al., 2022; Langston and Temme, 2019; Schanz and Montgomery, 2016), it does not overlap with the *d* exponents of the undisturbed valleys in our study area. The low values of the *d* exponent of the beheaded valleys, reflect a smaller increase of valley width *(log(W [m]))* per unit change in drainage area *(log(A [km$^2$]))* that is consistent with the process of beheading. During beheading, a valley loses its narrowest headwater sections, and consequently, the beheaded valley is wider at smaller drainage areas compared to undisturbed valleys (e.g., Fig. 3b). Farther downstream, as

drainage area increases through contribution from non-beheaded side tributaries, the effect of beheading on drainage area decreases and the valley width – area values become similar to those of undisturbed valleys (e.g., Fig. 3a-b). Additionally, the drainage area loss reduces the discharge and sediment transport capacity near the divide and may lead to aggradation, further widening the valley bottom (Brocard and van der Beek, 2006; Langston and Tucker, 2018) in small drainage areas. These effects act to lower the slope of the *log(W [m])* vs. *log(A [km$^2$])* regression line of

beheaded valleys compared to undisturbed valleys  (e.g., Fig. 3b).

The processes described above also increases the value of the $k_c$ coefficient of beheaded valleys compared to undisturbed valleys. However, whereas the median $k_c$ value and the overall $k_c$ range are indeed higher in beheaded valleys (Fig. 4 and Table 1), the $k_c$ difference is relatively small. The reason is likely that the $k_c$ coefficient reflects the valley width at a drainage area of 1 *km$^2$*. In the study area, a 1 *km$^2$* drainage area is reached only after the beheaded

section is joined by several undisturbed tributaries, which obscures the beheading influence, and blurs the difference in $k_c$ coefficient among the undisturbed and beheaded valleys.

    The negative value of the $d$ exponent for the reversed valleys (between -0.18 and -1) reflects downstream valley narrowing, supporting the inferred reversal of these valley sections (Harel et al., 2019). The three southern reversed valleys (10, 11 and 12 in Tables 1 and 2 and in Fig. 1d) yield $d$ exponents with an absolute value that is

similar to that of undisturbed valley sections. This observation is consistent with the view that (i) the geometry of the antecedent valleys whose flow direction was reversed is similar to that of the undisturbed valleys (e.g., Figs 2e, 3b and 6); and (ii) valley width did not drastically change following the reversal process. The $d$ exponents in the two northern valleys (8 and 9 in Tables 1 and 2 and in Fig. 1d) have higher absolute values, reflecting strong contrast between the narrow widths close to the knickpoint (several meters) and the anomalously high widths near the windgap

(>500m), that are likely associated with the E-W trending strike valley that accommodates the windgaps.

**5.2. Identifying reorganization from valley width - drainage area scaling**

The distinct ranges of $d$ exponent for the undisturbed and reorganized valley categories are consistent with the hypothesis that drainage reorganization modifies the scaling between valley width and drainage area. Based on these results, we suggest that such scaling differences could help to identify instances of drainage reorganization and point

to specific categories of reorganization according to the values of inferred $d$ exponents in these sections. Importantly, invoking $d$ exponents to support reorganization requires comparing the suspected reorganized valley sections to undisturbed sections with similar environmental conditions because the valley widening rate could be strongly affected by local factors. Among these factors are the lithology of the valley bed and walls  (Brocard and van der Beek, 2006; Langston and Temme, 2019; Langston and Tucker, 2018; Schanz and Montgomery, 2016), the climatic and glacial

history of the landscape (Chen, 2021; Clubb et al., 2022; Hancock and Anderson, 2002), and geologic structures activated by tectonic forcing (Keen-Zebert et al., 2017; Whittaker et al., 2007a). Consequently, local deviations of the $d$ exponent likely indicate reorganization only when the deviation is constrained across similar lithologic, climatic, and tectonic settings. When these conditions are met, we expect that deviations of the $d$ exponent can serve as an effective tool for identifying reorganized drainages, regardless of the lithology and climate conditions.


**5.3. Influence of channel slope on predictions of reversed valley width**

Our analysis reveals that Eq. (2), which includes the local slope, yields better valley width predictions for reversed valley sections compared to Eq. (1). This is evident from the high values of adjusted $R^2$ for the multivariate regression and the finding that in most cases, the best-fit area and slope exponents, (*b* and *c*, respectively) are of the same order of magnitude.

While the relation between valley width, drainage area and channel slope was postulated based on theoretical considerations (Brocard and van der Beek, 2006), the specific processes by which channel slope affects valley width remained vague. We suggest that in our study area, part of the correlation between the valley width and channel slope is linked to trends seen at the downstream edge of the reversed sections, above the knickpoint, where the valley narrows and the channel incises into the bedrock and steepens (e.g., valley transects in Fig 2e-f, and Fig. S9 in the Supplement). Narrowing and steepening close to the upper lip of the knickpoint is likely associated with flow acceleration above the knickpoint (Haviv et al., 2010) that forms a juvenile narrow valley (Fig. 2f). Deeper incision above the knickpoint may cause local bank collapse that erodes the remanent terraces of the paleo valley and establishes a narrower valley that amplifies the narrowing of reversed valleys towards the knickpoints (Fig. S9 in the Supplement) and may increase the absolute value of the exponent *d* (Eq. 1). This process likely reflects a transient response to reorganization and the onset of valley width adjustment to the new drainage direction.

Valley 10 (Tables 1 and 2 and in Fig. 1d) is an interesting exception in this context. Here, despite a ~80 m high knickpoint at the edge of the reversed section, valley narrowing above the knickpoint is not prominent, and slope increase is absent (Fig. S9 and S10 in the Supplement). The lack of incision above the knickpoint in valley 10 could imply a recent episode of knickpoint migration to its current location. Accordingly, in this case, the adjusted $R^2$ of the multivariate regression (Eq. 2, Table 2) is only slightly higher than the standard $R^2$ of Eq. (1) (Table 1), and the slope exponent is distinctively low (Table 2), suggesting that in this site, the inclusion of slope does not meaningfully improve the prediction of valley width.

**5.4 Timescales and mechanisms of valley and channel width adjustment in reversed drainages**

The comparison between the valley and channel width patterns in reversed valley 12 (Fig. 6) reveals a distinct contrast between the valleys' negative *b* exponent, reflecting a downstream valley narrowing, and the positive *b* exponent of

the channel, reflecting a downstream channel widening. We suggest that this field case demonstrates a temporal

snapshot, where the channel width is adjusted to the new drainage area distribution inflicted by the drainage reversal.

In contrast, the valley width is not yet adjusted to the change in drainage area (Fig. 6b), consistent with the longer

timescales expected for valley adjustment relative to the channel  (Hancock and Anderson, 2002; Langston and

Tucker, 2018).

In the reversed category, an increase in drainage area is associated with the process of gradual divide

migration within the antecedent valley (Harel et al., 2019). Field observations from valley 12, show that the latest

drainage area redistribution phase is set by a small avulsion in a colluvial fan that drains the northern flank of the

valley close to the windgap (Fig. S11 in the Supplement). The main active flow path of this fan flows east toward the

reversed section; however, an older path that drains westward toward the beheaded section is not completely

abandoned and is likely active when the main flow path is flooded (Shelef and Goren, 2021). This setting reflects a

recent episode of flow diversion and redistribution of discharge from the beheaded to the reversed valley. Therefore,

the inferred positive and high exponent of the channel width - drainage area scaling (Fig. 6) demonstrates rapid channel

adjustment, in line with previous studies that proposed rapid response of channel width to environmental changes

(Amos and Burbank, 2007; Attal et al., 2008; Morell et al., 2020; Snyder and Kammer, 2008; Yanites, 2018).

In deeply incised channels, where lateral erosion is minimal and hillslope erosion is enslaved to channel

incision, we suggest that the time required to erode the antecedent valley bottom, $t$ *[kyr]*, depends on  the channel's

vertical incision rate $E_v$ *[m/kyr]*, the averaged hillslope angle, $\emptyset$ *[m/m]*, and the width of the antecedent valley ($W$

*[m]).:*

$$t = \frac{W}{2E_v}\emptyset \tag{3}$$

We apply Eq. (3) to approximate the time required for reversed valley 12 to completely erode its antecedent valley.

Morphometric measurements in valley 12, based on the TanDEM-X DEM, yield a maximal valley width of 125 m

near the windgap and $\emptyset \sim= 0.4$. Based on the ages of abandoned terraces along channels of similar drainage areas and

climate (Enzel et al., 2012), we estimated $E_v$ to range between 0.5 to 0.05  *m/kyr*. With these values, Eq. (3) predicts

a time range of 50-500 *kyr*. However, the underlying assumption of Eq. (3),  that hillslopes respond instantaneously

to channel incision, is not necessarily valid in arid environments where hillslope processes are slow (Ben-Asher et al.,

2017; Dunne et al., 2016). The high slopes of the terrace flanks in valley 12, exceeding 0.4 in some cases, support a

delayed response of the hillslope to channel incision. We therefore suggest that the predictions of equation (3)
represent a lower bound when applied to arid environments.

### 5.5 Implications for landscape evolution

Delayed valley versus channel adjustment in response to reorganization (Fig. 6) and the diverging response of valleys

of different reorganization categories (Fig. 4) have important implications for landscape evolution. We explore an

example of such an implication by inspecting the influence of channel and valley widths adjustment on a proxy to the

unit stream power ($\omega = \rho g Q S/W$ [Watt/m], $\rho$: density, $g$: gravitational acceleration, $Q$: discharge), which is commonly

used for evaluating fluvial erosion rate (e.g., Harbor, 1998; Magilligan et al., 2015). Given that $\rho g$ can be treated as a

constant and that $Q$ is typically proportional to drainage area, the unit stream power is proportional to $p_{sp} = AS/W$ [m]

(Whittaker et al., 2007a). Using the width of the formative flows for $W$, $p_{sp}$ is calculated to explore changes in unit

stream power across the windgap between reversed valley 12 and beheaded valley 6 (Fig. 7a).

Field observations demonstrate that the formative flows of reversed valley 12 are currently confined to the

narrow, actively incising channel (Figs. 2e, 6, and 7a), resulting in comparably high values of $p_{sp}$. In contrast, across

the windgap, in beheaded valley 6, the $p_{sp}$ values are an order of magnitude lower because here, the wide valley defines

the width of the formative flows, which fully occupy the flat alluvial valley bed (Figs. 2d and 7a). This difference

results in a substantial step-change in the $p_{sp}$ values across the windgap (Fig.7b, black dots), suggesting that the

windgap is unstable and likely to migrate in the direction of beheaded valley 6.

Harel et al. (2019) proposed that in this study area, valley reversal initiates and extends by gradual windgap

migration along an antecedent valley. Windgap migration increases the drainage area along the reversed segment, and

according to the response documented in valley 12, contributes to the incision of a narrow channel within the wider

antecedent valley. Across the windgap, drainage area loss hinders incision on the beheaded side, and the formative

flow remains exceptionally wide. These differences in valley response and formative flow width contribute to the $p_{sp}$

step-change across the windgap, and, consequently, to erosion rate differences that promote further windgap migration

toward the beheaded valley. This 'width feedback' adds to the drainage area feedback ( Willett et al., 2014) in

facilitating  ongoing windgap migration, extending the reversed segment and shrinking the beheaded segment.

595         Importantly, the $p_{sp}$ values of the beheaded valley (valley 6) represent a conservative estimation. First, the flatness of the beheaded valley hints that transport-limited conditions and aggradation may dominate changes in valley bed elevation rather than vertical erosion (Brocard and van der Beek, 2006; Finnegan and Balco, 2013). Second, the limited sediment transport capacity and associated sediment aggradation over the beheaded valley bed could contribute to a relatively permeable valley fill that increases infiltration, and decreases the effective discharge per drainage area.

600         The step-change in $p_{sp}$ values across the windgap emphasizes the importance of accurate channel and valley width estimates when exploring the evolution of landscapes undergoing drainage reorganization. More specifically, when the width is approximated based on simple width-drainage area scaling, without accounting for the influence of reorganization (e.g., green open circles in Fig. 7b), the $p_{sp}$ values meaningfully deviate from the measured values (green open circles relative to black dots in Fig. 7b), the aforementioned step-change in $p_{sp}$ is not recognized (Fig. 7b,

green open circles), and the windgap will be wrongly assumed as stable (Fig. 7b). In contrast, $p_{sp}$ estimations based on scaling that account for reorganization are consistent with the measured $p_{sp}$ values (blue and pink rhombuses relative to black dots in Fig. 7b) and emphasize the erosion rate difference between the reversed and the beheaded valley sections that reflects the instability of the windgap between them.


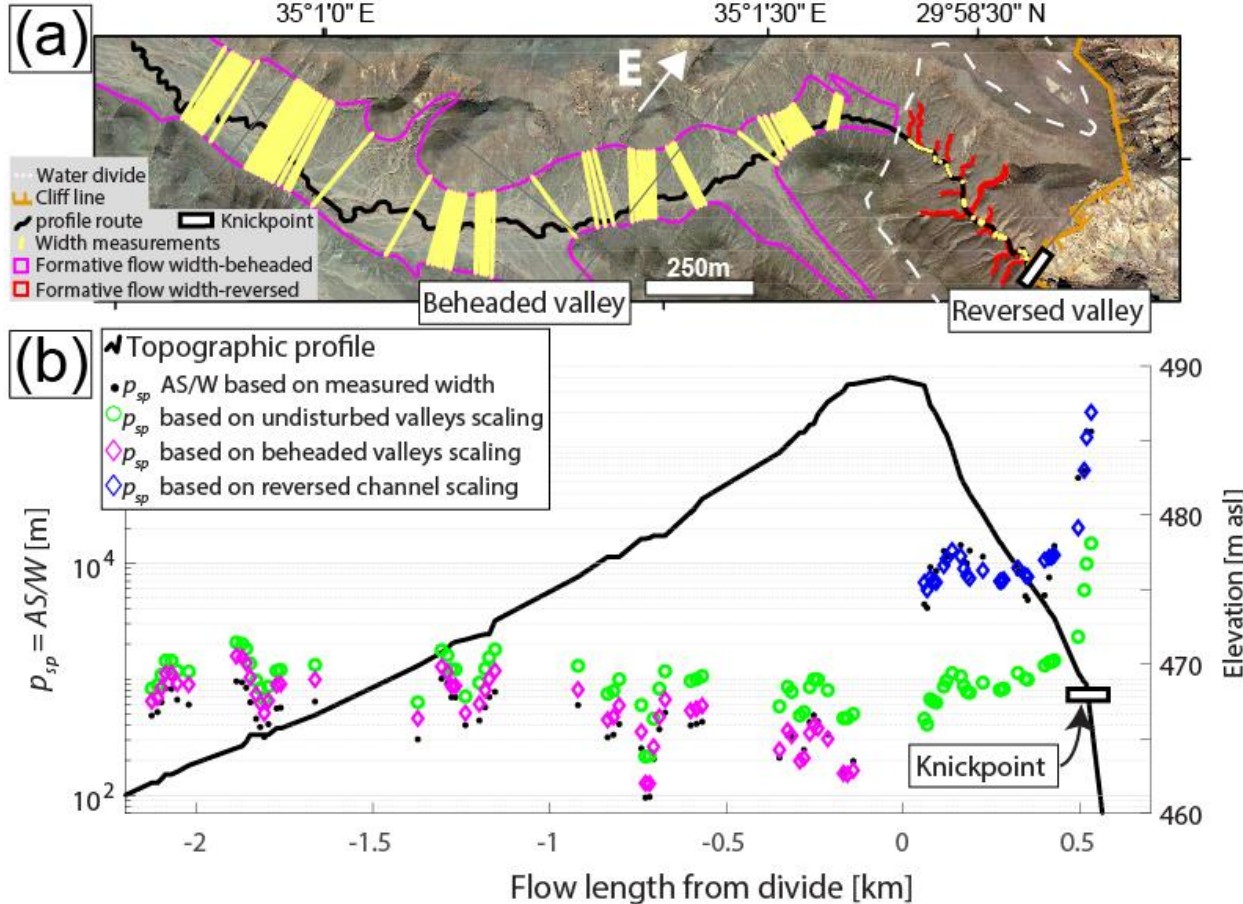

**Figure 7: A proxy for unit stream power ($p_{sp}$=AS/W) along a profile from a reversed valley (# 12) to a beheaded valley (# 6) across a windgap. (a) An orthophoto of the reversed and beheaded valley sections that share a common windgap. The black line marks the profile route that follows the main channels and crosses the divide. The dashed white line marks the divide, and yellow lines mark measured width transects of the formative flow width used for calculating $p_{sp}$. In the reversed valley, east of the windgap, the active drainage is confined to an incised and narrow channel (Fig. 6a), whose width is used for estimating $p_{sp}$. West of the windgap, in the beheaded side, the formative flow width aligns with that of the valley. (b) $p_{sp}$ estimates, based on different measurements of the formative flow width: i) Measured from DEMs (black dots), ii) Computed based on the median of the fitted predictors for undisturbed valleys in the study area, that is, without accounting for reorganization: W=100\*A$^{0.47}$ (green open circles), iii) Computed based on the scaling fitted for a beheaded valley (#6 in table 1): W=139\*A$^{0.18}$ (pink rhombuses) and iv) Computed based on the channel scaling for a reversed channel (Fig. 6b): W=7.6\*A$^{0.62}$ (blue rhombuses). The contrast between the morphological properties of the channels in the reversed and the beheaded valleys generates a distinct step-change in the $p_{sp}$ values across the windgap, which can promote continuous windgap migration. The trend is not predicted by $p_{sp}$ estimations that do not account for the unique width scaling in reorganizing valleys (green open circles).**

## 6. Conclusions

Analysis of undisturbed and reorganized valley sections in the Negev desert reveals that the scaling between valley width and drainage area (Eq. 1) is affected by drainage reorganization. Each reorganization category is associated with a distinct range of drainage area exponent values, *d,* that relate the valley width to the drainage area. In the undisturbed valleys, the range of *d* exponent values is overall consistent with values reported in previous studies (Attal et al., 2008; Kirby and Ouimet, 2011; Whittaker et al., 2007a). The *d* exponent values of the beheaded valleys

are positive and smaller than in undisturbed valleys. In the reversed valleys, the *d* exponent values are negative, reflecting valley narrowing with increasing drainage area. We propose that these deviations could benefit future studies that aim to identify and categorize drainage reorganization by comparing the width-area scaling of suspected reorganized drainages to those of undisturbed valleys with similar lithologic, climatic and tectonic conditions.

     Most reversed valleys exhibit a poor covariance between slope and drainage area. Therefore, in this category,

the valley width scaling was also inspected through Eq. 2, which incorporates both the slope and drainage area as predictors of valley width. This multivariate analysis results in higher adjusted $R^2$ values than those produced by Eq. (1) and resulted in negative exponents for both area and slope, (*b* anc *c,* respectively) with the same order of magnitude, indicating that they are both significant for the valley width prediction in the reversed category.

     In the reversed valleys, differences in width-area-slope scaling also occur between a channel and its hosting

valley. In a reversed valley section analyzed in detail, we found that the channel width is best fitted by a positive area exponent *b*, whereas the exponent for the valley width is negative, reflecting a faster adjustment of channel width to the post-reorganization drainage area distribution relative to the adjustment of valley width. This case study of a reversed valley section that shares a common windgap with a beheaded valley illustrates the significance of the contrasting timescales of channel and valley width adjustment for landscape evolution. The difference between the

narrow active channel in the reversed section and the wide formative flows that occupy the entire width of the beheaded valley across the windgap results in a step-change in the unit stream power across the windgap, used here as a proxy for fluvial erosion rate. Consequently, the step-change in unit stream power promotes divide migration and is a part of a divide migration feedback: erosion rate gradients across the windgap push the windgap toward the beheaded valley, which has a smaller unit stream power due to its wider channel and lower slope. Windgap migration

induces rapid channel width adjustment on the extending reversed side, while on the beheaded side, adjustment is delayed, sustaining the gradient in unit stream power. This feedback suggests that the differing response of channel

and valley width in different reorganization categories could maintain ongoing divide migration and may add to the slope and area feedbacks that were previously invoked as drivers of divide migration (Plant et al., 2014; Shelef and Goren, 2021; Willett et al., 2014). This width feedback could be easily overlooked if the channel width is

parameterized based on standard scaling relation, which is commonly assumed in large-scale landscape evolution models (e.g., Goren et al., 2014; Lague et al., 2014; Shobe et al., 2017; Yanites et al., 2013).

Insights from this study point to several venues for future research, for example: What are the constraints on the timescales over which the deviation in scaling persists? How do they vary with climate and lithology? Could the values of area exponents $d$ or $b$ quantify the temporal state of channel and valley adjustment? And what is the relation

between the dynamics and rates of divide migration to the width adjustment of valleys and channels?

## Data availability

The study is based on the copyrighted 12m TanDEM-X, that is available, with service fee charge, via the link: https://tandemx-science.dlr.de/cgi-bin/wcm.pl?page=TDM-Proposal-Submission-Procedure. The data of width, slope, and drainage area of the analyzed sections, and kml shapefiles of the valley bottom polygons before and after manual editing and of the width measurements, are available at: https://doi.org/10.5281/zenodo.6970603.

## Code availability

The ArcGIS model for the width measurements is fully described in the Supplement, and can be downloaded at: https://doi.org/10.5281/zenodo.7007928 .

## Supplement

The Supplement related to this article is available online at: ***

## Author contributions

LG and ES conceptualized the project. EH developed the software for extracting valley and channel width and performing the regression, with input from LG and ES. EH analyzed the data, generated the figures, and wrote the initial draft. LG, ES, and OC reviewed and edited the manuscript and supervised the research. HG introduced us to the field area and contributed to the ideas presented in this manuscript.

## Competing interests

The contact author has declared that neither they nor their co-authors have any competing interests.

## Disclaimer

Publisher's note: Copernicus Publications remains neutral with regard to jurisdictional claims in published maps and institutional affiliations.

## Acknowledgements

This study is supported by the National Science Foundation – Binational Science Foundation (NSF-BSF) under Grant Numbers 1946253 (NSF), and 2019656 (BSF). We thank the German Aerospace Center (DLR) for providing the 0.4arcsec tandem-X DEM. We also acknowledge our field assistants, Eitan Meidad, Gad Reifman, Haran Henig, Omri Porat, Tom Kaner, and Yaakov Prois. Elhanan Harel thanks the Nehemia Levtzion scholarship for its support. We thank Drs. Charles Shobe, George Hilley, and the anonymous reviewer for their constructive comments that significantly improved this Manuscript. Dr. George Hilley suggested Eq. (3) for estimating the time required to eliminate the antecedent valley width.

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
