# Peer review of "Drainage reorganization induces deviations in the scaling between valley width and drainage area"

_Earth Surface Dynamics, 2021_

## Author Comment (AC1)

Dear editors and reviewers,

We thank the reviewers for their insightful and constructive comments and revised the manuscript and analysis to address these comments. The data reanalysis took longer than initially planned, and we apologize for the delay in responding to the comments.

The revised manuscript includes a few main changes:

1. We now primarily rely on the equation $W = k_c A^d$ to quantify valley width scaling, rather than on the equation $W = k_b A^b S^c$ .
2. The slope-area covariance was analyzed for all valley sections and was found distinctively low in the reversed valley sections. We hence also applied the equation $W = k_b A^b S^c$ for these sections to explore the role of both slope (S) and drainage area (A) in such settings.
3. The method for slope calculation along the channel was changed, and is now based on a regression slope (i.e., elevation vs. distance) over a running window along the channel profile.

Specific replies to the reviewers' comments are given below; comments are in **bold** font, and answers are in regular font.

**Reply to review by Dr. Charles Shobe**

**In this contribution, the authors seek to test the hypothesis that drainage reorganization changes the scaling among drainage area, slope, and valley (and channel; more on that later) width. They use topographic data and field observations from an area undergoing drainage divide migration to show that beheaded channels have large, positive area-valley width exponents (width is large for a given area) and reversed channels have negative area exponents (width decreases with increasing drainage area) relative to channels they deem to be unaffected by piracy. Slope results are less clear, but the authors do find that reversed channels show different slope exponents between the channel, which has likely adjusted to the reversal, and the valley, which has not. Results suggest that area-width scaling might be useful for finding channels affected by divide migration, and also point to the fact that assumptions of instantaneous adjustment of width to area variations (i.e. those included in effectively all modern landscape evolution models) are to some extent wrong.**

This paper speaks to a couple of topics that are of great relevance to geomorphology right now, notably 1) the importance of drainage divide migration in shaping landscapes and 2) the need to improve our understanding of channel width dynamics if we want to be able to simulate landscape evolution. I found this paper a joy to read, and it is overall in good shape. It is mostly easy to follow, asks and answers an interesting question, and the results are largely well supported by the data. I do have a couple of conceptual/methodological concerns that I would like to see addressed before publication. I hope the authors find my suggestions helpful in improving what is already a very interesting piece of work.

Overall points:

1. **Writing and organization of the introduction: We as readers do not find out the knowledge gap that the authors intend to fill (aside from the abstract) until line ~115 or so, after a reasonably thorough discussion of channel width theory and drainage reorganization dynamics. I strongly suggest that before section 1.1, the authors make a statement similar to lines 116-119 stating that they are going to investigate width patterns in basins undergoing capture events. This could follow reasonably well from the existing lines 61-63. That way, readers will know the point of the paper before they slog through the width theory.**

   We appreciate your advice and added the following sentence before section 1.1: "The current study explores valley and channel width scaling in transient conditions that emerge from processes of drainage reorganization. "

2. **Channel width versus valley width: I know the authors are aware of this distinction, but it is a tricky issue. The classical width scaling ideas apply to river channels, but the width measurements that ultimately get made are of valley width (Figure 3). While I appreciate that Equation 2 is in some sense "general" (line 218), it does ultimately rest on the idea of open channel flow operating according to Manning's equation and having a constant bankfull width/depth ratio, whereas valley width is often conceptualized as being dependent on some measure of lateral channel mobility (e.g. Hancock and Anderson, 2002; Limaye and Lamb, 2014) which may or may not be**

**correlated with channel width. Given the massive size of these beautiful valleys (Figure 2), surely their cross-sections are not analogous to those of a channel in terms of being self-formed by some formative discharge. I do not suggest that the authors re-do their analysis to try to only use channel width, because that would be a lot of work and probably not possible in this intriguing landscape of broad-valleyed ephemeral channels. I would like to see however two things:**

**1) a dedicated, well-referenced subsection in the discussion clarifying for readers how the focus on valley width (as opposed to channel width) might influence the results of the study (for example, incorporating—or dismissing if you disagree—my comment above that valley width can in some cases be a function of channel mobility rather than width).**

Thank you for this constructive comment. The first issue raised in this comment is, "The classical width scaling ideas apply to river channels, but the width measurements that ultimately get made are of valley width." To address this, we revised the manuscript such that the primary analysis uses Eq. (1), $W = k_c A^d$ which, in addition to its more common application to channel width, was also widely used for the analysis of valley width (Beeson et al., 2018; Brocard and van der Beek, 2006; Clubb et al., 2022; Langston and Temme, 2019; Langston and Tucker, 2018; May et al., 2013; Schanz and Montgomery, 2016; Snyder et al., 2003; Tomkin et al., 2003). In revised section 1.1, we also expanded on the difference between the processes underlying the widening of channels and valleys- including the role of channel mobility in valley bottom formation. We note that to the best of our knowledge, the use of Eq. (1) for both valleys and channels relies mostly on empirical observations.

Another change in the revised manuscript is that Eq. (2), $W = k_b A^b S^c$, is used only for the reversed valley category, where the slope may influence the width independently of the drainage area (i.e., due to a low covariance between slope and area). We agree with your statement that Eq. (2) is generally based "on the idea of open channel flow operating according to Manning's equation and having a constant bankfull width/depth ratio." However, at the end of revised Sect. 1.2 we now cite Brocard and van der Beek, (2006), who provided a theoretical framework for relating valley width, drainage area, and slope in the form of Eq.

(2). The potential effect of channel slope on valley width is addressed in revised Sect. 1.2 and is further discussed in revised Sect. 5.3 which is entirely devoted to a potential effect of channel slope on valley width.

Regarding your comment about including a **"subsection.. clarifying.. how the focus on valley width (as opposed to channel width) might influence the study results**":

The findings of this study are predicated on the long timescales that are required for valleys to adjust to drainage area changes, that are mentioned in the revised manuscript, Sect. 1.3 : "For valleys, the time-gap between the change in drainage area and width adjustment is expected to be even longer (than channels), most likely in the order of tens of thousands of years (Hancock and Anderson, 2002; Langston and Tucker, 2018), because valley width represents the channel location integrated over long periods (Schumm and Ethridge, 1994; Tomkin et al., 2003)." We propose that the deviations in the valley width- area scaling that were found in the reorganized categories in the study area persist only due to the time-gap between the change in drainage area and the valley width adjustment. This was also explored in the comparison between the width scaling of the channel and the width scaling of the valley in reversed section 12 (Fig. 6 in the original and the revised manuscript) . We found that in contrast to the valley, which is not adjusted to the new drainage area distribution inflicted by the reorganization, the channel width is adjusted to the drainage area, indicating that the channel adjusts faster than the valley. Therefore, focusing on valley width increases the likelihood of the documented deviations that were observed in the drainage area exponent following reorganization.

**And 2) more careful word choice when describing the quantities measured in the study. Often just the term "width"is used; I know it might seem tedious but I do think it would be helpful to make sure it is always prefaced by either "channel"or "valley."**

Thank you for this important suggestion – Done in the revised manuscript.

3. **Slope data: Slope data extracted from DEMs is notoriously noisy (the classic discussion is Wobus et al., 2006, already cited by the authors). In this case the authors blame that noise for the strange near-complete lack of slope influence on width in all but the reversed valleys (line 429). I suspect that they are right, but this leads me to wonder about the slope measurement methodology. Of course slopes in these channels can be low as the authors note, and that makes things hard. But I have a few other ideas about methodological causes for this result.**

1. **Figure S5 brings up an interesting point, which is that the D8 flow path along which slope is measured may be substantially more sinuous (and therefore lower sloping) than the actual path "felt" by the water during a formative flow event. Is the flow path in Figure S5 actually showing a thalweg that is substantially enough incised into the valley that it would guide formative flows? Or would flow basically be straight down-valley (and therefore experience a much higher bed slope)? I could easily be wrong, but supposing that the latter case was true it might make more sense to simply use the valley centerline as the line along which slope is approximated.**

   Thank you for pointing this out. We agree that the DEM-based thalweg shown in Fig. S5 is likely more tortuous than the formative flow lines parallel to the valley centerline. In the revised manuscript, the background of Fig. S5 was replaced with an orthophoto to illustrate this. Notably, the discrepancy between the low slopes resulting from the DEM-based thalweg relative to formative flows is observed only in the flat, west-flowing valley sections (i.e., beheaded and undisturbed). In the reversed valleys, the incised channels commonly guide the formative flow.

   We appreciate your suggestion to change the slope calculation. However, in this study, using slope based on the DEM thalweg can be justified for the following reasons:

   1. The most straightforward method for applying a slope area analysis is to extract the drainage area and slope from the same pixel. Otherwise, when the slope is extracted from the centerline, it is not always clear what drainage area should be assigned to the pixel. The uncertainty

introduced by such drainage area assignment may be higher than that caused by potential deviations between the sinuous flow pathways and the path of formative flows.

2. As mentioned above, the difference between the thalweg based (i.e., sinuous) and formative slopes exists only in the west-flowing valleys, where the slope is used only for the slope-area analysis (Eq. (2) that necessitates slope is applied only to the east-flowing valleys where the slope follows a distinct channel thalweg). The slope-area results from the DEM-based thalweg show a rather strong slope-area covariance in the west flowing sections. We propose that even if higher slopes were found along the centerline, they would not significantly change the inferred covariance between the slope and area. Overall, we think the differences between these slopes are unlikely to meaningfully influence the results presented in the study.

**2. On a separate note, Wobus et al 2006 discuss log-bin averaging schemes for slope data to reduce the noise. Did the authors try log-bin averaging? If so and it didn't work, it would be good to say that in the paper. If not, it would be worth a shot to see if that changes the relationships between slope and width.**

Thanks for this constructive note. Following this suggestion, we performed log-binned slope-area analysis to quantify the covariance between slope and area along the valleys (Figure 1 here).

In the reversed category, the slope-area covariance was low ($R^2$ between 0.5-0.01), and therefore the channel slope was used in a multivariate regression that was performed to explore the width-area-slope scaling through Eq. (2). In this multivariate regression, log-binning was not applied because of two reasons:

1. Log binned averaging for a multivariate regression requires a 3D log binning, i.e., the data needs to be binned in both dimensions, of slope and drainage area. When we started the analysis presented in this study, we used a 3D log-binned approach. We ended up not including it in the paper's results because it was a relatively complex procedure that did not necessarily enhance the regression quality.

2. Most of the reveresd valleys have relatively small amount of width measurements (between 12 and 50, mean 27.2). In such cases, log-bin averaging creates bins with only one or few data points per bin, resulting in unreliable mean values of the bins, which make the log-binning meaningless.

[Figure]

**Figure. 1: Best-fit correlations of the slope-area power-law of all valley sections. The regressions were applied to log-binned data (black squares) to reduce the natural noise within the slope data (blue dots).**

**One last thought: calculating the slope between any two DEM points is always an error-prone thing given the vertical error in DEMs. What about a linear regression along the five pixels where the best-fit slope then gets assigned to the middle pixel? This must be less sensitive to the error in any individual pixel.**

Thank you for this valuable advice. In the revised manuscript, the slope is extracted through a regression.

**My concrete suggestion here is that the authors make sure that their somewhat surprising results with respect to slope are not simply a function of methodology. This could be done by testing other slope calculation and averaging methods, or by explaining very clearly why other methods failed or weren't tried.**

Thanks. In the revised manuscript, the slope is extracted through a linear regression (main comment #3), and the slope-area analysis uses log-binning (main comment #2). As you rightly expected, the new slope extraction methods meaningfully reduced the noisy slopes in the west-flowing valleys, which you defined as "somewhat surprising results with respect to slope". Following the new slope results, we eliminated the discussion section based on the scattered slope observation.

Line comments:

**34: "control river dynamics".**

Done in the revised manuscript.

**37: "and hydrologic modeling".**

Done in the revised manuscript.

**64: As detailed in main comment 1, this is a good place to say how your study contributes to understanding of width under transient conditions.**

Done in the revised manuscript.

**65: Consider being a bit more specific in the header: "channel and valley width scaling"or similar.**

Changed to "Width- area scaling in channels and valleys."

**70 and other equations: the multiplication symbol is not needed.**

Done in the revised manuscript. Thanks.

**83: Consider including a brief half-sentence explanation for why landslides influence width scaling. It will not be obvious to everyone I think.**

Done in the revised manuscript.

**86: specify rock uplift?**

Changed to "rock uplift rate" in the revised manuscript.

**97: see line 70 comment.**

Done in the revised manuscript.

**112: I think there is more certainty here than your language implies. I suggest changing to "width adjustment does not occur...".**

Thanks, the text was rephrased in this sentence and in other places in the revised manuscript.

**113: Can you be more specific about what constitutes a "long" timescale?**

In Sect. 1.3 the revised manuscript, we explicitly mention the expected order of magnitude for timescales required for the adjustment of channels and valleys: "Studies that measured channel widths in drainages that experienced recent anthropogenic drainage area perturbations reported ongoing width variations that prevailed for several decades (e.g., Jones, 2018; Snyder and Kammer, 2008). Based on a theoretical model, Turowski (2020) postulated that the timescale of channel width adjustment to discharge perturbations is in the order of thousands of years. For valleys, the time-gap between the change in drainage area and width adjustment is expected to be even longer, most likely in the order of tens of thousands of years (Hancock and Anderson, 2002; Langston

and Tucker, 2018), because valley width represents the channel location integrated over long periods (Schumm and Ethridge, 1994; Tomkin et al., 2003)".

**120: This is one example of an issue that crops up repeatedly. Which type of width, channel or valley? If both, state that explicitly.**

The valley. Text was fixed in this sentence and in other places in the revised manuscript.

**127-131: This is an excellent statement of the work's contribution!**

Thanks!

**152: can delete "had"**

Done in the revised manuscript.

**Figure 1: The level of zoom in (a) is awkward: not close enough to see distinct geomorphic features, and not far enough out for unfamiliar readers to know where in the world they are. I suggest adding another panel for a true "location figure," which could then allow (a) to be zoomed in farther if you want.**

Thank you for pointing this out. Following your suggestion, panel (a) is now a generalized location map, and panel (b) is a zoom-in map that emphasizes the main structures in the area, namely the rift-like morphology and the highlands that bound it (Figure 2 here).

[Figure]

**Figure 2.** (a) Orientation map with coastlines (blue) showing the study area location (red star). (b) Shaded elevation map, illustrating the regional rift morphology along the plate boundary (dashed white line) adjacent to the study area (black rectangle).

**Figure 1: (b) is a beautiful illustration; very nice work.**

Thank you!

**Figure 1: looking at (c) brings up a question for me: how undisturbed are the "undisturbed" basins? Surely they must be experiencing some drainage area loss to escarpment migration. I agree with the authors that the data show diagnostic differences between those streams and the beheaded and reversed ones, but it might be worth adding a few sentences assessing the extent to which any stream draining this divide can be considered "undisturbed"since likely it just means losing drainage area at some much slower rate than the beheaded streams.**

This comment was addressed by describing the undisturbed valleys more precisely in revised Sect. 2.2: "In some cases, field evidence indicates that some portions of the drainage area along the low-relief interfluves were lost due to divide migration

associated with cliff receding. Yet, the receding cliff does not intersect the incised portion of the valleys; therefore, these valleys are referred to as 'undisturbed'."

**172: can delete "a".**

Done in the revised manuscript.

**178: there is some vague terminology here: "relatively rapid" and "slower." Can these be made any more specific?**

We rephrased the text and provided a general rate in the description in revised Sect. 2.2: "Farther downstream, typically at a distance less than 1 km from the valley head, the valley bed becomes alluviated, and its width increases to tens of meters."

**183: maybe "indicated" instead of "recognized?"**

Done in the revised manuscript.

**215-219: see major comment 2. This is where I think you need a much more detailed justification for extrapolating a relation based on open-channel flow mechanics from the channel scale to the valley scale.**

In addition to our answer to major comment 2, the justification for using both Eq. (1), $W = k_c A^d$ and Eq. (2), $W = k_b A^b S^c$ for valleys is embedded in the introduction of the revised manuscript:

In Sect. 1.1 we present Eq. (1) for channels suggesting that although it is commonly based on empirical observations, it is consistent with a recent study that modeled channel width based on a mechanistic approach (Turowski, 2020). Later in that section, we present the mechanisms controlling valley width and explicitly write, "Despite the different processes that underlie the widening of channels and valleys, empirical observations suggest that the relation between the valley width and drainage area follows a similar power-law scaling to Eq. (1)." That is, although some research has investigated the channel width scaling through a mechanistic approach, the common usage of the equation $W = k_c A^d$ is overall empirically based, either for channels or valleys.

Later in Sect. 1.2, we present Eq. (2) that was previously applied only to channels. To relate this equation to valley width, we refer to the work of Brocard and van der Beek

(2006), who provided a theoretical framing for the usage of Eq. (2) for valley width, although this wasn't explored in the field. At the end of this section, we present ways by which channel slope could influence valley width; for example, channel steepness can serve as a proxy for lithological variations that set the mode of valley widening at different reaches.

**235-244: This procedure necessarily contains some steps that could be considered subjective (e.g., manual polygon editing, etc). That is understandable. But when paired with the authors' statement of "data available upon request," it makes it hard to view this study as easily reproducible. I don't think the journal requires it (though they should!), but I would strongly encourage the authors to archive all the GIS files relevant to their valley width measurements in a Figshare repository or similar. Then I would be less concerned about the fact that polygons had, for example, to be manually edited.**

We agree that the data processing must be reproducible. As you recommended, we added to the resubmitted manuscript the KML polygons of the valley bottom before & after the manual edits, accompanied by the valley width measurements. Unfortunately, any publishment of the raw TanDEM-X DEM data is strictly prohibited; therefore, the DEM or any of its direct derivations cannot be shared in the Supplement.

**245: I have always written "thalweg," but the EGU typesetting staff will sort it out one way or the other.**

This specific line was rephrased, but in other places we changed to "thalweg".

**255: See my major comment 3 on slope extraction. In short, I am not sure that this is the best procedure (endpoint differencing along the flow line that is not binned/averaged in any way).**

The slope extraction method was changed as suggested in major comment #3 and the text was modified respectively in Sect. 3.1 in the revised manuscript: "The channel slope used for exploring slope-area relations and channel and valley width predictions following Eq. (2) was estimated along the flow network (thalweg) by using the slope of a linear regression between elevation and distance over a centered 7-pixel running window."

**270: Again the lack of any binning worries me given the recommendations of Wobus et al (2006). There could be, at a minimum, some justification of why none was done.**

We applied the log-binning to the slope-area analysis (Figure 1), but not when using the slope in the multivariate regressions of Eq. (2) applied to the reversed valleys. Please see our answer to major comment #2, where we provide the justifications for this.

**300: I also recommend reporting the DEM versus dGPS measurement differences in % in addition to m.**

Done in the revised manuscript.

**330-334: In line with major comment 3, I worry that this result could be partially due to methodology. I could certainly be wrong, but it is at least worth a bit more of a detailed treatment in the discussion.**

Thank you. This comment refers to results that were excluded from the revised manuscript, because currently Eq. (2) does not apply to the undisturbed and beheaded valleys.

**Figure 5: An easy way to make this more readable would be to label the exponent x-axis labels with the quantity they correspond to, e.g. "b exponent (area)"**

Done in the revised manuscript.

**351: "Farther," not "further".**

Done in the revised manuscript.

**354: "for an additional".**

Done in the revised manuscript.

**Figure 6: The valley bottom outline does not look very convincing in the context of the air photo because it is hard to visualize the breaks in slope that should delineate it. Consider replacing the air photo with a DEM as the background on this image, or add another panel with the DEM.**

Thanks. To address this issue, we added contour lines with intervals of 2m that were bolded each 10m, which easily reflect the slope break through the density of the contour lines (figure 3).

[Figure]

**Figure 3. (a) Valley (blue) and channel (orange) polygons and width transects, sketched over a 0.5 m resolution orthophoto. The slope break between the valley bottom and the hillslope is emphasized by the density change of the 2m contour lines (thin black lines).**

**373: you undersell your results! Consider replacing "could be" with "are".**

Done in the revised manuscript.

**374: "could be" -> "can"**

This text was excluded from the revised manuscript.

**453: can we get any more specific than "relatively rapidly?"**

The specific text you commented on does not appear in the revised manuscript. Sect. 1.3 in the revised manuscript provides general channel and valley adjustment timescales from the literature: "Studies that measured channel widths in drainages that experienced recent anthropogenic drainage area perturbations reported ongoing width variations that prevailed for several decades (e.g., Jones, 2018; Snyder and Kammer, 2008). Based on a theoretical model, Turowski (2020) postulated that the channel width adjustment timescale to discharge perturbations is in the order of thousands of years."

In the new Sect. 5.4, also devoted to adjustment timescales, we discuss the adjustment rates of the channel in response to a recent reorganization event. However, absolute rates are beyond the scope of the present study. In the framework of a different research, we are dating the abandonment ages of terraces in valley 12. We propose that the results from the dating, which we plan to complete soon, will assist in associating the channel and valley adjustment processes to a temporal framework.

**473: Again related to my point about channel versus valley width: if there are places where the two are especially similar, it seems like we should be prepared to understand differences that might occur in the places where the two widths are NOT similar. This is one of the points I'd like to see added in a discussion of this issue.**

A meaningful portion of the discussion in the revised manuscript is now devoted to the reversed category, where channel and valley width differ. Revised Sect. 5.3 shows how the channel incision above the knickpoint is part of the dynamic adjustment of the valley. Revised Sect. 5.4 discusses the adjustment timescales of the channel width vs. the valley width, and revised Sect. 5.5 highlights why the decoupling between channel and valley in the reversed valleys is consequential for landscape evolution.

**535: You could mention specifically here that this simplified approach is what is used in effectively all large-scale LEMs. One of the key exciting results of your study to me is a path forward for breaking down some of these oversimplistic assumptions regarding concordance between area and width.**

Thanks, these lines in the revised manuscript now read: "This width feedback could be easily overlooked if the channel width is parameterized based on standard scaling relation, which is commonly assumed in large-scale landscape evolution models (e.g., Goren et al., 2014; Lague et al., 2014; Shobe et al., 2017; Yanites et al., 2013)."

**Reply to review by Dr. George Hilley**

**This contribution uses field observations, TanDEM-X-derived DEMs, and high-resolution SfM (and GNSS) field surveys to study how valley and channel width vary with drainage area for watersheds experiencing drainage reorganization. To do this, they use a series of 12 watersheds, some of which have been previously documented, in the Negev desert. Here, a rift-like structure has produced a vertical offset on the order of 500 m, which cuts across the heads of a series of (formerly) west-flowing drainages. This offset creates a knickpoint, which redirects flow within a portion of the formerly west-flowing channels to the east. As the knickpoints expand, the divides migrate. However, incision has not been sufficient to erode a valley-depth of the landscape, and so the valley geometries of the former flow direction are preserved within this landscape. The authors exploit this fortuitous circumstance to measure how the headward migration of the drainage divide produces channel width - area relations at odds with the valley width - area relations encoded in the former regime. This, with some scaling arguments used to calculate stream power in the channels, suggests that divides may be more mobile than might appear due to a simple stream-power rule because of the narrowing of channel width that accompanies incision of the east-flowing channels.**

General Comments:

**1) This is a very nicely executed study in a fascinating natural system. The fact that the old valley network can be clearly delineated and measured is fortuitous for addressing this problem.**

Thank you!

**2) The study is predicated on the idea that valley width scales as a function of watershed area in a way that is similar to channel width. While this may empirically be the case, the valley width (in fluvial environments) may be set by a different set of factors (e.g., erodibility of the bank materials that scales migration rate, human modification of channel systems such as mill ponds, landslide damming of rivers) than the channel width. I'm fine with an empirical measurement of valley width to demonstrate that scaling (for instance, the Beeson work), and these different factors are acknowledged. But, it could be**

**valuable to dedicate a bit more description to this distinction – that one can be related, somewhat directly to the flows that traverse the channel, while the other will be related to the evolution and migration of the channels over time.**

Thank you for this constructive comment. In the revised manuscript, Sections 1.1 and 1.2 now emphasize the difference between the factors that govern channel width and valley width. We also explicitly write that both equations are generally based on empirical observations. Below are more details regarding how this comment is addressed through the text structure of the revised manuscript:

In Sect. 1.1 we present the equation $W = k_c A^d$ for channels and suggest that although it is commonly based on empirical observations, it is consistent with a recent mechanistic-based approach to channel width modeling (Turowski, 2020). We then discuss different mechanisms that shape valley width and explicitly write: "Despite the different processes that underlie the widening of channels and valleys, empirical observations suggest that the relation between the valley width and drainage area follows a similar power-law scaling to Eq. (1) (Beeson et al., 2018; Brocard and van der Beek, 2006; Clubb et al., 2022; Langston and Temme, 2019; Langston and Tucker, 2018; May et al., 2013; Schanz and Montgomery, 2016; Snyder et al., 2003; Tomkin et al., 2003) ".

Later, in Sect. 1.2, the equation $W = k_b A^b S^c$ is presented (Eq. (2)). This equation was previously applied in the field only to channels. To relate this equation to valley width, we cite Brocard and van der Beek (2006), who provided a theoretical framing for using Eq. (2) to describe valley width scaling, although this relationship was not tested in any field setting. At the end of this section, we present ways by which channel slope could influence valley width; for example, that channel steepness can serve as a proxy for lithological variations that set the mode of valley widening at different reaches.

**3) One contribution of this work is to present a novel method of extracting valley width, which is useful for a number of types of studies as the citations of the authors indicate. It seems to work particularly well for a class of valley morphologies in which the valley floor can clearly be distinguished from the valley side-slopes by assigning a (calibrated) slope threshold. It's worth noting that this is not generally the case, and so this method will fail when there is a gradation between valley floor and side-slopes.**

Thank you for suggesting this. Importantly, we note that valley width extraction in our study is done in two stages, first where the valley bottom polygon is generated and second, where width is measured in optimal locations. While the second part is based on an algorithm developed through this study and comprises a contribution of this work, the first part of generating the valley bottom polygon is based on the existing algorithm of Gilbert et al. (2016). Indeed, this method mostly fits valley morphologies characterized by a distinct slope break, which is common in the study area. To address your comment, we added a sentence in Sect. 3.2 in the revised manuscript that reads: "This method particularly suites valley morphologies where the valley bottom can be easily distinguished from the valley walls based on a distinct slope break which is the case in most of the studied valley sections."

**Also, it's probably worth noting that the valley width in this case is the width of the valley floor.**

Agreed, thank you. In the revised manuscript, we added to the beginning of the section describing the valley extraction: "In the undisturbed and beheaded categories, the valley width refers to the flat valley bottom.." .

**Finally, I would suspect that the method works best when the valley and channel sinuosity are similar. Otherwise, might a sinuous channel lead to aberrant projections normal to the trend of a sinuous inner channel to the VBET? These issues might be discussed a bit more in the methods section.**

The valley width extraction method is not affected by channel sinuosity. After generating the valley bottom polygon, the valley width transects are projected perpendicular to the valley bottom centerline, regardless of the channel geometry. The full procedure of the width extraction is described in detail in Section S1 and Figs. S1-S5 in the Supplement of the original and the revised manuscripts. To eliminate potential confusion related to this, we edited the last paragraph in Sect. 3.2 in the revised manuscript. Particularly, the sentence "The algorithm identifies points along the valley thalweg" was replaced by "The algorithm identifies points along the valley centerline".

**4) The three classes of watersheds, "Undisturbed", "Beheaded", and "Reversed", seem to imply a conception of how water is routed through this landscape. In particular, if one were to simply consider the DEM, probably all of the watersheds would be "Beheaded" or "Reversed", since the escarpment cuts across the**

**apparent heads of the former drainages. But, it seems from Figure 1b that the network is incised into a low-relief surface. Figure 1b thus implies that the distinction between a beheaded and undisturbed channel rests on whether or not the incised portion of the drainage has been cross-cut by the escarpment / other channels. Maybe I am confused about this, but if this is indeed the case, then a channel is undisturbed only if water is sourced exclusively from the incised portion of the landscape. If this is the case, I think the watershed areas must be calculated based on those incised areas of the landscape, rather than the entire DEM.**

We agree with your observation that some of the valleys defined as "Undisturbed" are losing drainage along their low-relief, unincised interfluve as an outcome of cliff migration. However, the extent to which these valleys are losing drainage area is relatively small, and we prefer to maintain their definition as "Undisturbed" for abbreviation. To address this comment, we revised their definition, which now reads: "In some cases, field evidence indicates that small drainage areas along the flat interfluves were lost due to divide migration associated with the cliff receding. Yet, the incised part of the valleys is not affected by the cliff line; therefore, these valleys are referred to as 'undisturbed.'

**5) The idea that channel widths adjust more quickly than valley widths seems intuitive. Thus, in a landscape with such clear divide migration, one would expect channels widths to increase with area, while the valley widths might behave in an opposite manner, which is I think the main point of the paper. A couple of thoughts on this:**

**A) I think the reversed valley width scaling can persist for no longer than the valley width divided by the incision rate * average hillslope angle. The signature could be much shorter, but this would be the maximum amount of time that would be required to remove the former valley morphology.**

Thank you for sharing this thought with us. Your idea was embedded in the second part of Sect. 5.4 in the revised manuscript, where the maximal time required to remove the antecedent valley width is given by the equation $t = \frac{W}{2E_v}\emptyset$. This section provides an estimated range for 't', and later discusses the suitability of this equation to the specific

conditions in the study area. In the acknowledgment section, we explicitly thank you for suggesting this approach.

**B) I was not completely on board with the association of the valley and channel widths in the eastward-flowing channels for the stream-power calculation (and the inferences that flow from this). The authors assert that in the westward-flowing channels, there are a series of anastomosing low relief-channels that occupy virtually the entire valley width. I think this is being shown in Figure 7, where the width measurements for the stream power calculations are the yellow "width measurements"? Yet, the photograph of the valley seems to indicate that only a fraction of those valleys are occupied by a channel. If this is the case, the width effect on the stream power calculation might be somewhat overstated.**

You raise here an essential point we had rethought about, after reading your comment and comments from the other reviewers. In the revised manuscript, we clarify this vagueness by referring to the width of formative flow (instead of the width of the channel) in calculating a proxy for stream power ($p_{sp}$). Our field observations indictate that during significant rain events, the braided channels in the west-flowing valleys are flooded such that flows occupy the entire valley width (as illustrated in Fig. 2d in the original and the revised manuscripts). Therefore, in these valleys the formative flow width and the valley width are equal. In contrast, the formative flow in the reversed valleys is limited to the narrow width of the channel, which is incised into the paleo-valley so the flow is constrained by the flanks of remanent valley terraces.

This idea is manifested in the text of revised Sect. 5.5: "Field observations demonstrate that the formative flows of reversed valley 12 are currently confined to the narrow, actively incising channel, resulting in comparably high values of $p_{sp}$. In contrast, across the windgap, in beheaded valley 6, the $p_{sp}$ values are an order of magnitude lower because here, the wide valley defines the width of the formative flows, which fully occupy the flat alluvial valley bed."

**C) Another way to have a larger contrast in stream power between the reversed and beheaded channels might be to have increased infiltration and transient storage of precipitation in the valley alluvium on the low-sloped, eastward flowing channels relative to their west-flowing counterparts. I don't know much about the flashiness of the Negev desert, but some alluviated valleys in California produce surface runoff only in very large storms because of the effect of infiltration into**

**the low-sloped valleys. This might stand in contrast to the steeper-sloped segments that form as the drainage divide migrates. Perhaps this is worth some discussion as well?**

Thank you. Such hydrologic mechanisms can indeed affect the feedbacks that drive the divide migration process in the study area. We added this factor in Sect. 5.5 in the revised manuscript, where we discuss the 'width feedback' between the divide migration process and the rapid channel width adjustment in the reversed valleys: "Importantly, the $p_{sp}$ values of the beheaded valley (valley 6) represent a conservative estimation... the limited sediment transport capacity and associated sediment aggradation over the beheaded valley bed could contribute to a relatively permeable valley fill that increases infiltration, and decreases the effective discharge per drainage area."

**Thanks very much for the opportunity to be part of this work. The field site is a real gem, which has been exploited well by the proponents. I am support of its acceptance to ESurf, pending consideration of some of the issues I mentioned above.**

**George Hilley.**

**This manuscript explores the relationships among drainage area, slope, and valley width in three types of drainage basins in various states of basin reorganization: undisturbed basins, beheaded basins, and reversed basins. The authors find that the values of the exponent on drainage area distinguish among the three basin types, and a significant, negative exponent on slope is present in reversed basins. I thought this manuscript was well written and nicely organized. I think this manuscript needs minor revisions prior to publication. The main issue I think the authors need to address is increasing the impact or value of the findings from this study, largely in the discussion section. Below I describe several sections in the manuscript that I think should be expanded and/or discussed in more detail. At the end of this review, I have some minor line-by-line comments.**

**1. The authors find unique values of the drainage area exponent b in the three valley types and mention that the different valley types 'could' be characterized by the value of the exponent b (lines 319 – 323 and 373 - 374). But the authors later point out (lines 407 - 411) that there are several other reasons one might find distinct values of b, even in neighboring basins (e.g. Schanz, et al., 2016). What distinguishes the characteristic values for the b exponent in these reorganized basins from different b values due to differences in lithology (Schanz et al., 2016; Langston and Temme, 2019; Brocard and van der Beek, 2006)? What I would like to see here is a more thorough description of when or under what conditions reorganized drainage basins can be identified by the value of the b exponent. This is mentioned but needs more discussion. Further discussion of when variation in the b exponent may point to drainage basin reorganization will help to guide future readers to using the results presented here properly and increase the impact of the paper.**

We appreciate this advice. A well-referenced list of factors that influence the range of the b exponent is first mentioned in Sect. 1.1 in the revised introduction: "The exponent's range was mostly attributed to differences in channel bank properties, where more erodible and/or fractured banks widen faster than resistant and intact banks (Spotila et al., 2015; Whitbread et al., 2015; Wohl and Achyuthan, 2002; Wohl and David, 2008). Other studies invoked climatic variations and anthropogenic disturbances to explain

variations in the d exponent (Bertrand and Liébault, 2019; Faustini et al., 2009; Snyder et al., 2003). "

To further address your comment, we added a new subsection, no. 5.2, titled: 'Identifying reorganization from valley width and drainage area scaling'. This section expands on how to conduct a similar analysis in other field sites, mainly emphasizing the importance of comparing the reorganized drainages to undisturbed sections with similar climatic, lithologic, and tectonic conditions.

**2. Also related to impact of the paper, would it be worthwhile for future researchers to extract valley width, drainage areas, and slopes from basins with an unknown reorganization history to attempt to identify new locations with past drainage basin reorganization? Perhaps not given the difficulty of extracting valley bottom width; therefore, it seems like the key takeaways of this paper are the differences in unit stream power across drainage divides and evidence for different time scales of channel and valley width adjustment.**

We partly agree with the statements here. As to the question, "**would it be worthwhile for future researchers to extract valley width, drainage areas, and slopes from basins with an unknown reorganization history to attempt to identify new locations with past drainage basin reorganization?**" our answer is yes, we do think that deviations in valley-width-drainage area scaling can identify reorganization, under specific limitations given in revised Sect. 5.2 (or in the original Sect. 5.1), even without a-priori knowledge of the drainage history. Nonetheless, in revised Sect. 5.2 we write that "scaling differences could serve as supportive evidence for drainage reorganization." We prefer a conservative phrasing because this method was tested only in one field case. We look forward to future studies that will use scaling deviations in other field cases and assist in establishing the method we suggest here.

Regarding "**the difficulty of extracting valley bottom width**"; valley bottom extracting is becoming simpler nowadays thanks to the ongoing development of extraction methods (e.g., in this paper or Clubb et al., 2022). We think that in studies that attempt to validate / eliminate potential reorganization scenarios, analysis of valley width may be worth the effort involved in the extraction process.

**3. Figure 6 that shows increasing channel width going downstream but decreasing valley width going downstream is especially interesting. Does this pattern exist in the other reversed drainage basins?**

The negative drainage area exponents for the valley width- drainage area scaling indicate that downstream valley narrowing occurs in all reversed sections. We did not determine the width-drainage area-slope relations for all of the channels in the reversed category (other than channel 12), because such analysis requires a high-resolution DEM, that is available only for valley 12. We refer to this shortly in the Sect. 4.4 in the revised manuscript, that describes the observations in valley 12 : "Field observations show that while the reversed valley narrows downstream (i.e., eastward), the channel width increases in the downstream direction (Fig. 6a). A similar pattern was also observed in other reversed valleys.". In more detail, valley 10 is the most prominent case where the downstream increasing channel width pattern is apparent based on a 0.5m orthophoto and Google Earth images. The other reversed valleys, i.e., 8, 9, and 11, also exhibit a general channel width increase; however, a robust examination of the channel width-area relations in all the reversed valley sections will require a separate analysis as well as acquiring high-resolution topographic data for all these valleys.

**Can the authors give us any idea on the timescales of adjustment for channel width vs. valley width? The authors start to discuss this on lines 458 – 461, but don't say anything more on timescales besides "longer" vs. short timescales for valleys and channels respectively. I understand the authors might not have the data to give a hard number for the timescale of adjustment, but a more in depth discussion of what factors play into the timescale of adjustment would also be helpful.**

Thank you for this comment. In the revised manuscript, we referred to the timescales issue more extensively. In Sect. 1.3 in the revised manuscript we cite studies that estimate the order of magnitude of timescales required for channel and valley width adjustment following drainage area change: "Studies that measured channel widths in drainages that experienced recent anthropogenic drainage area perturbations reported ongoing width variations that prevailed for several decades (e.g., Jones, 2018; Snyder and Kammer, 2008). Based on a theoretical model, Turowski (2020) postulated that the timescale of channel width adjustment to discharge perturbations is in the order of

thousands of years. For valleys, the time-gap between the change in drainage area and width adjustment is expected to be even longer, most likely in the order of tens of thousands of years (Hancock and Anderson, 2002; Langston and Tucker, 2018), because valley width represents the channel location integrated over long periods (Schumm and Ethridge, 1994; Tomkin et al., 2003). "

In Sect. 5.4 in the revised manuscript we now provide information about channel and valley adjustment timescales available without absolute dating. We point to field observations showing that the channel width responds to the addition of drainage area that was redirected to the reversed valley, likely by a recent avulsion event. This evidence, and the adjusted width of the downstream widening channel, indicate that the channel width likely adjusts rapidly to drainage area changes. In the second part of Sect. 5.4 in the revised manuscript, we suggest a quantitive estimation for the maximal time it takes to remove the antecedent valley bottom, based on the ratio between the initial width and the vertical erosion rates, a contribution suggested by reviewer 2 (Dr. Hilley). With this equation, we calculate a range for the maximal adjustment time of the valley near the divide, accompanied by a discussion of the equations' suitability in the study area.

Additionally, we are currently working on luminescence dating of terraces in valley 12, which are expected to shed new light on the dynamics of the channel and valley development in the context of windgap migration. Results will be published in a separate contribution.

**4. I would also like to see a discussion of how the hyper-arid climate plays into these findings. How would these findings be different in a humid climate? Discussion on this point would add to the impact of the manuscript and findings.**

We appreciate this comment. Climate is one of the essential factors that influence the exponent $d$, as we mentioned in the revised manuscript: "Other studies invoked climatic variations and anthropogenic disturbances to explain variations in the d exponent (Bertrand and Liébault, 2019; Faustini et al., 2009; Snyder et al., 2003)." In this context, we preferred to refer to climate, generally, rather than hyper-arid particularly because the accurate influence of hyper-arid climate on the width-area scaling of channels and valleys is not yet well determined (Chen, 2021; Tan et al., 2021; Wolman and Gerson, 1978).

This study has two main contributions to understanding the aridity effect on channel\valley width scaling:

1. The exponents of the valley-width drainage area power law in the hyper-arid undisturbed valley sections are consistent with the exponents reported by previous studies conducted in humid climates, as we write in Sect. 5.1 in the revised manuscript: "In our study area, the $d$ exponent values of the undisturbed valleys range between 0.26 and 0.54, consistent with previously published exponent values for valleys (Beeson et al., 2018; Brocard and van der Beek, 2006; Clubb et al., 2022; Langston and Temme, 2019; Schanz and Montgomery, 2016; Shepherd et al., 2013; Snyder et al., 2003; Tomkin et al., 2003)."

2. The hyper-arid climate is suspected of playing a role in the unique observation in valley 12, where channel width is adjusted to the new hydrological conditions and the valley width is not. In the second part of Sect. 5.4 in the revised manuscript, after estimating a range for the maximal time remove the antecedent valley bottom, we note that, "the underlying assumption of Eq. (3), that hillslopes respond instantaneously to channel incision, is not necessarily valid in arid environments where hillslope processes are slow (Ben-Asher et al., 2017; Dunne et al., 2016). The high slopes of the terrace flanks in valley 12, exceeding 0.4 in some cases, support a delayed response of the hillslope to channel incision. We therefore suggest that the predictions of equation (3) represent a lower bound when applied to arid environments."

Importantly, even in cases where the exponent values differ between climates, we expect the distinction between positive $d$ values in beheaded and undisturbed valleys to negative values in reversed valleys to persist across climates.

As a concluding remark, we note field observations from a study that we currently perform in reversed drainages in the humid Appalachian mountains, where field observations differ from those in the Negev desert. The study area in Appalachia generally lacks evidence of downstream narrowing of the valleys, and fill terraces that assist in distinguishing the channel from the valley (e.g., valley 12 in Fig. 6 in the original and the revised manuscript) are rarely preserved. Although these observations are likely related to climate, they could also occur due to the time that had passed since the reversal, that was sufficient for the drainage to adjust to the new drainage area. However, these are preliminary observations, and we look forward to proceeding and investigating this topic in future research.

**Line by Line comments:**

**Line 50 – 51: is this section discussing channel width or valley width? I found that in several places in this manuscript, I was not sure which one the authors were referring to. See also lines 120 – 121; 471**

In the case of these lines, we referred to both valley and channel width. To avoid similar confusions, the text in the revised manuscript clarifies whether it refers to channel, valley, or both in each place where width is mentioned.

**Line 179 – 181: Nice definition of valley vs. channel, thank you.**

Thank you!

**Line 232 – 234 and Fig. 6a: outlining a polygon representing the valley bottom can be difficult. In figure 6a, the area inside the polygon doesn't look like a valley bottom to me, with many tributaries carved into the defined valley bottom and no visible break in slope that distinguishes the valley bottom from the sides of the valley. Perhaps this polygon was created based on a slope map of the area, with a threshold slope dividing valley bottom and valley wall. But I can't see this in figure 6a. Would be helpful to see a figure that shows how this valley bottom polygon was drawn.**

Thank you. We added contour lines that illustrate the slope break, which defines the outlines for the valley bottom polygon (please see Figure 3 here in our response to Dr. Shobe.)

**Figure 3: Is Figure 3d showing data from valley 10 and the fit from valley 11? Or is this a typo?**

Thank you for noticing this; it was a typo. In the revised manuscript, the data of valley 11 is shown using two different equations: first as an example of W-A scaling that was analyzed in all valleys, and is shown togheter with the scaling in valleys 1 and 5 (Figure 4a here), and second as an example of W-A-S scaling that was analyzed only in the reversed valleys (Figure 4b here).

[Figure]

**Figure 4:** **(a) Linear regression fitted lines from log-transformed valley width and drainage area, for the undisturbed, beheaded, and reversed valleys 1, 5, and 11, respectively. The dashed lines represent 95% confidence bounds. The equations in the bottom right are the linear models' $k_c$ coefficients and d exponents. (b) Multivariate regression results with the associated $k_b$ coefficient and b and c exponents for the reversed valley 11. The 95% confidence interval is represented by error bars.**

**Additionally, this figure only shows W-A plots for three of the 12 valleys in this study. I suggest adding similar figures of all valleys to the manuscript or in the supplemental material. This could be 1 figure with 3 panels showing all data for normal, beheaded, and reversed valleys.**

Thank you for this suggestion, the W-A plots from all valley sections now appear in the Supplement of the revised manuscript (see Figure 5 here).

[Figure]

**Figure 5: Best-fits values for valley width-drainage area power law. Blue dots are the raw data, and solid and dashed black lines represent the least-square linear fit and 95% confidence bounds, respectively.**

**Line 322 – 323: valley width decreases going downstream. Does this necessarily mean a reversed valley? What else is needed to make that determination?**

Reversal was not established based on the W-A scaling, we report on the W-A scaling from the valleys that were a-priori categorized as reversed. The initial categorization was based on field evidence for drainage reversal, as we write in Sect. 2.2 in the revised manuscript: "Harel et al. (2019) identified these sections as reversed drainages based on the presence of barbed tributaries and west grading terraces that record the antecedent valley gradient, which is opposite to the present-day channel's drainage direction. The reversed valley sections share windgaps with beheaded valleys, indicating that they were part of an antecedent west-flowing drainage (Harel et al., 2019)."

**Line 324 – 325: Clarify that slope exponent values are non-unique only for normal and beheaded valleys. This is not entirely clear until readers reach lines 332 – 334.**

Thank you for this comment. This section was modified in the revised manuscript, where W-A-S scaling analysis is applied only to the reversed category. However, we were careful to eliminate similar unclear phrasing when describing the results in the revised manuscript.

**Figure 5: In my opinion, not a great figure. Can't see what the values of b and c are. Can only see that they are between 0 and 1 and either close to or far from 0 or 1.**

Thank you for pointing this out. To better identify the exponent values, we added minor grid lines to the x-axis in the figures illustrating the exponents of equations (1) and (2) in the revised manuscript.

**Line 388: Clarify. This makes me think that the authors are talking about channel slope, not slope of the W vs. Area line.**

In the revised manuscript these lines now read: "These effects act to lower the slope of the log(W [m]) vs. log(A [km$^2$]) regression line of beheaded valleys compared to undisturbed valleys."

**Line 404 – 405: yes, scaling differences between adjacent drainages is potentially evidence for drainage basin reorganization, but under what conditions? Scaling differences can also be lithology dependent could represent changes in uplift rate across catchments.**

Agreed. In Sect. 5.2 in the revised manuscript, we specify the restrictions required before comparing width-area scaling in reorganized and non-reorganized drainages.

**Line 433; 435 – 436: Does "slope exponents" refer to channel slopes or valley slopes?**

These lines were omitted from the revised manuscript.

**Line 448 – 449: A reference to a figure that shows this exception for valley 10 would be helpful.**

Figures S9 and S10 in the revised version of the Supplement (Figures 6 and 7 here) relate to this comment. Figure 6 compares slope and width variations with increasing drainage area in the reversed valleys, emphasizing the anomalous observation in valley 10, where

unlike the other valleys, slope decreases before approaching the knickpoint. Figure 7 illustrates the valley bottom polygon and width measurements on an orthophoto of valley 10, and presents the longitudinal profile and a plot that illustrates the slope and width variations vs. drainage area.

[Figure]

**Figure 6 : Slope and valley width (red and blue dots, respectively), plotted against drainage area at the reversed valleys.**

[Figure]

**Figure 7.** Analysis of the valley width and channel slope in reversed valley 10. (a) Valley bottom polygon, valley transects, and flow-pathway of valley 10, on an orthophoto (0.5m resolution). (b) Elevation profile of valley 10. (c) Valley width (blue) and channel slope (red) vs. drainage area. Here, width and slope do not covary near the knickpoint, unlike the trend of the other reversed valleys in the study area.

**Line 459 – 461, Figure 6 caption: Channel width increasing downstream while valley width decreases downstream is a cool finding. Do you see this in any other of the reversed valleys?**

All reversed sections in the study demonstrate a downstream decrease in valley width. As we answered in major comment #3, the channel width-drainage area scaling was measured only in valley 12. This is mentioned in Sect. 4.4 in the revised manuscript : "Field observations show that while the reversed valley narrows downstream (i.e.,

eastward), the channel width increases in the downstream direction. A similar pattern was observed also in other reveresd valleys.".

**What more can you say about this? What else can we get out of this finding?**

The contrast between the timescales of the valley and channel width adjustment is now more broadly discussed in Sect. 5.4 in the revised manuscript. In this section, we provide field observations to describe the erosive response to events of drainage area addition and to constrain the time it takes to erode the antecedent valley bottom through a simple theoretical model. In Sect. 5.5 in the revised manuscript, we further discuss in detail how the difference between valley and channel width patterns results in different erosion rates across the divide, which act on the landscape evolution by promoting feedbacks that enhance the divide migration process.

**Line 493: here's explanation of blue dots on figure 7. This explanation should be in fig 7 caption.**

Thank you for pointing at this. The blue dots in Fig. 7b (In the original manuscript) were replaced with open green circles in the revised manuscript. The caption of the figure now reads: "$p_{sp}$ estimates, based on different measurements of the formative flow width: … ii) Computed based on the median of the fitted predictors for undisturbed valleys in the study area, that is, without accounting for reorganization: $W=100*A^{0.47}$ (green open circles)".

**Figure 7: Why show channel bottom polygon in reversed and valley bottom polygon in beheaded?**

The channel bottom polygon in Fig. 7a (In the original and revised manuscripts) represents the formative channel width used to calculate the stream power proxy $p_{sp}$. In the revised manuscript, we changed the legend of polygons in Fig. 7a, which are now defined as "formative flow width - beheaded" and "formative flow width - reversed". The difference in the formative width of the channel is now described in Sect. 5.5 in the revised manuscript:  "Field observations demonstrate that the formative flows of reversed valley 12 are currently confined to the narrow, actively incising channel, resulting in comparably high values of $p_{sp}$. In contrast, across the windgap, in beheaded valley 6, the $p_{sp}$ values are an order of magnitude lower because here, the wide valley defines the width of the formative flows, which fully occupy the flat alluvial valley bed."

**Why calculate three different ways west of wind gap? which one is right/best?**

Fig. 7b  (In the original and revised manuscripts)  emphasizes two observations:

1. The step-change of the measured $p_{sp}$ (black dots in the revised version) across the divide).
2. The discrepancy between $p_{sp}$ estimates when $W$ is calculated based on scaling relations that consider the deviations in W-A due to reorganization (blue and pink rhombuses for reversed and beheaded valleys, respectively), vs. W computed based on scaling relations that do not consider the deviations. One outcome of this study is that width prediction that does not account for reorganization might overlook or underestimate the step-change in psp, as we write in the end of Sect. 5.5 in the revised manuscript: "When the width is approximated based on simple width-drainage area scaling, without accounting for the influence of reorganization (e.g., green open circles in Fig. 7b), the $p_{sp}$ values meaningfully deviate from the measured values (green open circles relative to black dots in Fig. 7b), the aforementioned step-change in $p_{sp}$ is not recognized (Fig. 7b, green open circles), and the windgap will be wrongly assumed as stable (Fig. 7b). In contrast, $p_{sp}$ estimations based on scaling that account for reorganization are consistent with the measured $p_{sp}$ values  (blue and pink rhombuses relative to black dots in Fig. 7b) and emphasize the erosion rate difference between the reversed and the beheaded valley sections".

**Line 506, fig 7 caption: should be red rhombuses, not blue?**

Yes. Thank you for noticing this. As part of the improvements of the revised manuscript, we changed the color of the symbols in Fig. 7b for consistency with other figures in the revised manuscript. Thus, green, pink, and blue colors were given to undisturbed, beheaded, and reversed valley sections, respectively.

**Line 518 – 519: "exploited with caution". What does that mean? Say more about how this can help identify and categorize drainage basin reorganization.**

We rephrased these lines that now read: "We propose that these deviations could benefit future studies that aim to identify and categorize drainage reorganization by comparing the width-area scaling of suspected reorganized drainages to those of undisturbed valleys with similar lithologic, climatic and tectonic conditions ".

**Line 536 – 537: This is a very interesting thought. I would like to see the authors take a stab at answering these questions, or at least share their thoughts on how climate and lithology might affect timescales of deviant scaling.**

Thank you. These questions were primarily posed to encourage future studies to use our findings and compare them to scaling relations in other cases of drainage reorganization.

However, some of the text additions in the revised manuscript relate to these questions. Regarding the effect of climate, please see our response to your comment no. 4. In the revised manuscript, we also added Sect. 5.4, titled "Timescales and mechanisms of valley and channel width adjustment in reversed drainages", that directly addresses the question in the end of the conclusion section: "What are the constraints on the timescales over which the deviation in scaling persists?." We are currently in the process of dating the terraces in valley 12, which will allow us to propose quantitative answers to this question that will be reserved for a future manuscript.

Answering these questions in more detail will require significant data gathering from various reorganized basins, which is beyond the scope of the current manuscript.

**References**

Beeson, H. W., Flitcroft, R. L., Fonstad, M. A. and Roering, J. J.: Deep-Seated Landslides Drive Variability in Valley Width and Increase Connectivity of Salmon Habitat in the Oregon Coast Range, JAWRA J. Am. Water Resour. Assoc., 54(6), 1325–1340, 2018.

Ben-Asher, M., Haviv, I., Roering, J. J. and Crouvi, O.: The influence of climate and microclimate (aspect) on soil creep efficiency: Cinder cone morphology and evolution along the eastern Mediterranean Golan Heights, Earth Surf. Process. Landforms, 42(15), 2649–2662, doi:https://doi.org/10.1002/esp.4214, 2017.

Bertrand, M. and Liébault, F.: Active channel width as a proxy of sediment supply from mining sites in New Caledonia, Earth Surf. Process. Landforms, 44(1), 67–76, doi:https://doi.org/10.1002/esp.4478, 2019.

Brocard, G. Y. and van der Beek, P. A.: Influence of incision rate, rock strength, and bedload supply on bedrock river gradients and valley-flat widths: Field-based evidence and calibrations from western Alpine rivers (southeast France), S. D. Willett al., Spec. Pap. Geol. Soc. Am, 398, 101–126, 2006.

Chen, A.: Climatic controls on drainage basin hydrology and topographic evolution, University of Bristol., 2021.

Clubb, F. J., Weir, E. F. and Mudd, S. M.: Continuous measurements of valley floor width in mountainous landscapes, , (February), 1–27, 2022.

Dunne, T., Malmon, D. V and Dunne, K. B. J.: Limits on the morphogenetic role of rain splash transport in hillslope evolution, J. Geophys. Res. Earth Surf., 121(3), 609–622, doi:https://doi.org/10.1002/2015JF003737, 2016.

Faustini, J. M., Kaufmann, P. R. and Herlihy, A. T.: Downstream variation in bankfull width of wadeable streams across the conterminous United States, Geomorphology, 108(3–4), 292–311, doi:10.1016/J.GEOMORPH.2009.02.005, 2009.

Goren, L., Willett, S. D., Herman, F. and Braun, J.: Coupled numerical-analytical approach to landscape evolution modeling, Earth Surf. Process. Landforms, 39(4), 522–545, doi:10.1002/esp.3514, 2014.

Hancock, G. S. and Anderson, R. S.: Numerical modeling of fluvial strath-terrace formation in response to oscillating climate, , (9), 1131–1142, 2002.

Harel, E., Goren, L., Shelef, E. and Ginat, H.: Drainage reversal toward cliffs induced by lateral lithologic differences, Geology, 2019.

Jones, J. C.: Historical channel change caused by a century of flow alteration on Sixth

Water Creek and Diamond Fork River, UT, Utah State University., 2018.

Lague, D.: The stream power river incision model: Evidence, theory and beyond, Earth Surf. Process. Landforms, 39(1), 38–61, doi:10.1002/esp.3462, 2014.

Langston, A. L. and Temme, A. J. A. M.: Impacts of Lithologically Controlled Mechanisms on Downstream Bedrock Valley Widening, Geophys. Res. Lett., 46(21), 12056–12064, 2019.

Langston, A. L. and Tucker, G. E.: Developing and exploring a theory for the lateral erosion of bedrock channels for use in landscape evolution models, Earth Surf. Dyn., 6(1), 1–27, doi:10.5194/esurf-6-1-2018, 2018.

May, C., Roering, J., Eaton, L. S. and Burnett, K. M.: Controls on valley width in mountainous landscapes : The role of landsliding and implications for salmonid habitat ABSTRACT, , 503–506, doi:10.1130/G33979.1, 2013.

Schanz, S. A. and Montgomery, D. R.: Geomorphology Lithologic controls on valley width and strath terrace formation, Geomorphology, 258, 58–68, doi:10.1016/j.geomorph.2016.01.015, 2016.

Schumm, S. A. and Ethridge, F. G.: Origin, evolution and morphology of fluvial valleys, 1994.

Shobe, C. M., Tucker, G. E. and Barnhart, K. R.: The SPACE 1 . 0 model : a Landlab component for 2-D calculation of sediment transport , bedrock erosion , and landscape evolution, , 4577–4604, 2017.

Snyder, N. P. and Kammer, L. L.: Dynamic adjustments in channel width in response to a forced diversion : Gower Gulch , Death Valley National Park , California, , (2), 187–190, doi:10.1130/G24217A.1, 2008.

Snyder, N. P., Whipple, K. X., Tucker, G. E. and Merritts, D. J.: Channel response to tectonic forcing: Field analysis of stream morphology and hydrology in the Mendocino triple junction region, northern California, Geomorphology, 53(1–2), 97–127, doi:10.1016/S0169-555X(02)00349-5, 2003.

Spotila, J. A., Moskey, K. A. and Prince, P. S.: Geomorphology Geologic controls on bedrock channel width in large , slowly-eroding catchments : Case study of the New River in eastern North America, Geomorphology, 230, 51–63, doi:10.1016/j.geomorph.2014.11.004, 2015.

Tan, C., Feng, S., Zhao, X., Shan, X. and Feng, S.: Longitudinal variations in channel morphology of an ephemeral stream from upland to lowland, Daihai Lake basin, North China, Geomorphology, 372, 107450, doi:10.1016/j.geomorph.2020.107450, 2021.

Tomkin, J. H., Brandon, M. T., Pazzaglia, F. J., Barbour, J. R. and Willett, S. D.: Quantitative testing of bedrock incision models for the Clearwater River, NW Washington State, J. Geophys. Res. Solid Earth, 108(B6), doi:10.1029/2001jb000862, 2003.

Turowski, J. M.: Mass balance, grade, and adjustment timescales in bedrock channels, Earth Surf. Dyn., 8(1), 103–122, doi:10.5194/esurf-8-103-2020, 2020.

Whitbread, K., Jansen, J., Bishop, P. and Attal, M.: Substrate, sediment, and slope controls on bedrock channel geometry in postglacial streams, J. Geophys. Res. Earth Surf., 120(5), 779–798, doi:10.1002/2014JF003295, 2015.

Wohl, E. and Achyuthan, H.: Substrate influences on incised-channel morphology, J. Geol., 110(1), 115–120, 2002.

Wohl, E. and David, G. C. L.: Consistency of scaling relations among bedrock and alluvial channels, J. Geophys. Res. Earth Surf., 113(4), 1–16, doi:10.1029/2008JF000989, 2008.

Wolman, M. G. and Gerson, R.: Relative scales of time and effectiveness of climate in watershed geomorphology, Earth Surf. Process., 3(2), 189–208, doi:10.1002/esp.3290030207, 1978.

Yanites, B. J., Ehlers, T. A., Becker, J. K., Schnellmann, M. and Heuberger, S.: High magnitude and rapid incision from river capture: Rhine River, Switzerland, J. Geophys. Res. Earth Surf., 118(2), 1060–1084, doi:10.1002/jgrf.20056, 2013.

---

## Author Response (AR2)

Dear editor,

Thank you for reading our revised manuscript and for your important comments. As you recommended, a link to the Arc-GIS model files was added in the "Code availability" section of the corrected manuscript. Below are the replies to the comments relating to the technical corrections; comments are in **bold** font, and answers are in regular font.

Sincerely,

Elhanan Harel, on behalf of the authors

**Abstract Line 25: Can you add "W" into the equation on the left hand side to make this clearer?**

Done.

**Line 65: I think you could make the end of the introduction stronger by stating that although tectonic, climatic and lithologic influences on channel width have been explored, there has been little work on the influence of drainage organisation on lateral erosion, and so this is what you set out to do.**

Thank you, we added this to lines 67-61 that now read: "While the influence of tectonic, climatic, and lithologic changes on valley and channel width has been extensively explored (e.g., Allen et al., 2013; Keen-Zebert et al., 2017; Marcotte et al., 2021), the effects of drainage reorganization, which imposes drainage area transiency, were mostly overlooked. The current study targets these effects by exploring valley and channel width scaling under transient conditions that emerge from processes of drainage reorganization."

**Lines 107 and 124: typo, should be "power law"**

Fixed, thank you.

**Line 159: remove apostrophe from valleys'**

Done.

**Line 569: Section 5.5: Change section heading to "Implications for landscape evolution"**

Done.

**Line 662: I struggled to understand this sentence: do you mean "rates of valley and channel width adjustment"?**

We meant rates of divide migration. Accordingly, the phrasing was changed to "And what is the relation between the dynamics and rates of divide migration to the width adjustment of valleys and channels?" (lines 664-665 in the corrected manuscript).

**Acknowledgements: typo, should be "George" rather than "Goerge"**

Thank you. Fixed.